# CoBo: Collaborative Learning via Bilevel Optimization

**Diba Hashemi**
EPFL
diba.hashemi@epfl.ch

**Lie He** [*]
Tencent Inc.
liam.he15@gmail.com

**Martin Jaggi**
EPFL
martin.jaggi@epfl.ch

## Abstract

Collaborative learning is an important tool to train multiple clients more effectively by enabling communication among clients. Identifying helpful clients, however, presents challenging and often introduces significant overhead. In this paper, we model *client-selection* and *model-training* as two interconnected optimization problems, proposing a novel bilevel optimization problem for collaborative learning. We introduce CoBo, a *scalable* and *elastic*, SGD-type alternating optimization algorithm that efficiently addresses these problem with theoretical convergence guarantees. Empirically, CoBo achieves superior performance, surpassing popular personalization algorithms by 9.3% in accuracy on a task with high heterogeneity, involving datasets distributed among 80 clients.[2]

## 1 Introduction

In a classic collaborative learning scenario, $n$ clients, each with a distinct machine learning task, seek solutions that potentially outperform their individual solvers through a collective effort. Common collaborative learning frameworks generally alternate between training local models on individual datasets and synchronizing updates among collaborators. More concretely, during the computation step, client $i \in [n]$ trains a $d$-dimensional model $\boldsymbol{x}_i \in \mathbb{R}^d$ to minimize its loss function, $f_i : \mathbb{R}^d \to \mathbb{R}$. In the subsequent communication step, client $i$ exchanges updates with clients, potentially benefiting from collaboration.

Despite the plethora of collaborative learning frameworks, the ideal approach to collaborate remains under-exploited. The FEDAVG [28, 18] algorithm learns a single global model over pooled datasets from all clients, i.e., $\min_{\boldsymbol{x} \in \mathbb{R}^d} \frac{1}{n} \sum_{i=1}^{n} f_i(\boldsymbol{x})$. However, due to heterogeneous data distributions among clients, a global model may significantly underperform compared to personal models trained on local datasets for certain clients, which can discourage their participation in collaborative training [29]. DITTO addresses this issue by training personal models with a regularization term that penalizes deviations from a global model [24]. Although DITTO enables personal models to leverage the global model, it offers only a coarse-grained level of collaboration. In instances where clients' data exhibit significant differences, the DITTO algorithm is constrained to facilitating collaboration at a global level, thereby neglecting the inherent heterogeneity among clients.

Clustering-based federated learning algorithms have been developed to accommodate scenarios in which clients' data originate from multiple clusters [11, 39]. Nevertheless, these algorithms typically inherit the limitations associated with clustering techniques, including the need to predetermine the number of clusters, initialize cluster centers, and other such prerequisites, which can diminish their practical utility.

---

[*]Corresponding author.
[2]The code is available at: https://github.com/epfml/CoBo.

38th Conference on Neural Information Processing Systems (NeurIPS 2024).

In this paper, we propose a bilevel optimization framework to enhance collaborative learning by discovering better structural relationships among clients. The inner problem focuses on optimizing a binary collaborator selection variable $w_{ij} \in \{0, 1\}$, determined based on a gradient alignment measure for each pair of clients. In the outer problem, we train personalized models $\boldsymbol{X} \in \mathbb{R}^{n \times d}$, incorporating a penalization term that accounts for the distances between clients, as dictated by the collaboration weights established in the inner problem.

The contributions of this paper can be summarized as follows:

- We model collaborative learning through a novel bilevel optimization formulation that yields more generalizable solutions by fully exploiting the inherent structure of collaboration.

- We propose COBO, an SGD-type alternating optimization algorithm that efficiently solves the bilevel problem. COBO scales with the number of clients $n$ and is elastic to the number of clients.

- We prove that COBO enjoys theoretical convergence guarantees for collaborative learning with cluster structures.

- Empirically, COBO surpasses popular personalized federated learning baselines in experiments involving highly heterogeneous federated learning settings and Large Language Models (LLMs).

## 2  Problem formulation

In this paper, we model collaborative learning as a bilevel optimization problem, where personalized models $\boldsymbol{X} \in \mathbb{R}^{d \times n}$ are trained in the outer problem, and collaborative weights $\boldsymbol{W} \in \mathbb{R}^{n \times n}$ are determined by the inner problem. More concretely,

$$\min_{[\boldsymbol{x}_1, \ldots, \boldsymbol{x}_n] \in \mathbb{R}^{d \times n}} \sum_{i=1}^{n} f_i(\boldsymbol{x}_i) + \frac{\rho}{2} \sum_{1 \leq i < j \leq n} w_{ij}^{\star} \|\boldsymbol{x}_i - \boldsymbol{x}_j\|_2^2 \qquad \text{(Model-Training)}$$

$$\text{where } w_{ij}^{\star} \in \underset{w_{ij} \in [0,1]}{\arg\max} \; w_{ij} \left\langle \nabla f_i \left( \frac{\boldsymbol{x}_i + \boldsymbol{x}_j}{2} \right), \nabla f_j \left( \frac{\boldsymbol{x}_i + \boldsymbol{x}_j}{2} \right) \right\rangle \quad \forall \, i, j \in [n],$$

$$\text{(Client-Selection)}$$

where $\rho > 0$ is a hyperparameter for penalization. We break down the formulation as follows.

**Outer problem: training personalized models.**   In the outer problem (Model-Training), client $i$ trains its model $\boldsymbol{x}_i$ by minimizing its loss function $f_i$, along with penalizing its distances to neighboring models, e.g. $\boldsymbol{x}_j$, as weighted by $w_{ij}^{\star} > 0$.

Our formulation is similar to DITTO [24], but with two key differences: First, DITTO uses uniform and fixed collaboration weights and penalizes the distance between $\boldsymbol{x}_i$ and a global model, whereas we penalize the distances between pairs of clients and adjust the collaboration weights during training. Consequently, when client tasks are heterogeneous—such as clients drawn from clusters—the performance of a global model deteriorates, and DITTO's local models cannot benefit from fine-grained collaboration. In contrast, our method is able to exploit such structure and achieve better performance in diverse settings.

**Inner Problem: Finding Collaborators.**   In the inner problem, we decompose the task of optimizing $\boldsymbol{W} \in \mathbb{R}^{n \times n}$ into independent sub-problems, one for each entry of $\boldsymbol{W}$. We relax the binary collaborator selection variable $w_{ij} \in \{0, 1\}$ to a continuous weight $w_{ij} \in [0, 1]$. As the objective function is linear with respect to $w_{ij}$, and the domain is convex, optimization algorithms such as Frank-Wolfe [10, 17] or projected gradient descent can efficiently find the maximizers, which occur at 0 or 1.

It is important to note that $w_{ij}^{\star}$ does not imply a permanent connection between clients $i$ and $j$, but rather a temporary assessment based on the current states of $\boldsymbol{x}_i$ and $\boldsymbol{x}_j$.

A simple inner problem with two clients is illustrated in Figure 1. The $f_1, f_2$ are their loss functions, and suppose $\boldsymbol{\mu}_1, \boldsymbol{\mu}_2$, and $\boldsymbol{\mu}$ are the minimizers of $f_1, f_2$, and $\frac{1}{2}(f_1 + f_2)$. Suppose $\boldsymbol{\mu}_1, \boldsymbol{\mu}_2$, and $\boldsymbol{\mu}$ are the minimizers of $f_1, f_2$, and $\frac{1}{2}(f_1 + f_2)$ respectively. The model weights at points $A, B, C$ demonstrate three scenarios for updating $\boldsymbol{W}$.

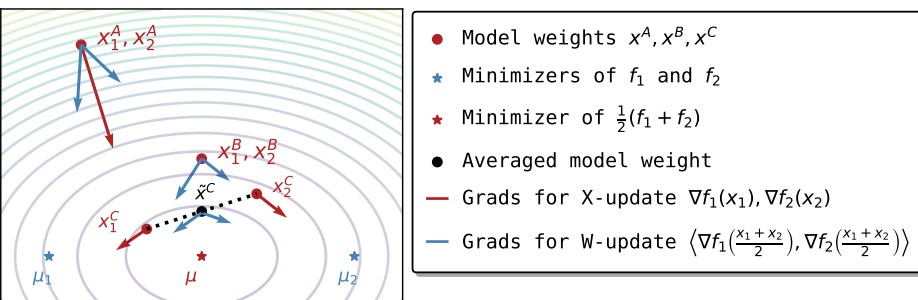

Figure 1: Diagram of the inner problem (Client-Selection) represented through a contour of $\frac{1}{2}(f_1+f_2)$. The blue arrows $\rightarrow$ are gradients computed at middle point $\frac{1}{2}(\boldsymbol{x}_1 + \boldsymbol{x}_2)$ to determine connectivity. The red arrows $\rightarrow$ represent gradients computed at local models to update model weights.

- **Point A:** The model $\boldsymbol{x}^A$ is far away from $\boldsymbol{\mu}$, i.e., $\|\boldsymbol{x}^A - \boldsymbol{\mu}\| >> \max_i\|\boldsymbol{\mu}_i - \boldsymbol{\mu}\|$. The descent directions of the clients have a positive inner product; therefore $w_{12} = 1$. Collaboration at this stage speeds up training.

- **Point B:** The model $\boldsymbol{x}^B$ is closer to $\boldsymbol{\mu}$, i.e., $\|\boldsymbol{x}^B - \boldsymbol{\mu}\| \sim \|\boldsymbol{\mu}_i - \boldsymbol{\mu}\|$. In this case, moving closer to the minimizer $\boldsymbol{\mu}$ of $\frac{1}{2}(f_1 + f_2)$ no longer helps both clients get closer to the minimizers of their own losses $\boldsymbol{\mu}_i$. The inner problem yields $w_{12} = 0$ and the clients disconnect.

- **Point C:** The models $\boldsymbol{x}_1^C$ and $\boldsymbol{x}_2^C$ are already disconnected. The gradients computed at their midpoint suggest they should remain disconnected; thus $w_{12} = 0$.

Because collaboration weights in Client-Selection are determined in a pairwise fashion, our formulation, unlike clustering-based methods [11, 39], does not require knowledge of cluster sizes, allowing clients to join and leave during collaborative training. This elasticity enables our method to be applicable in a wider range of scenarios.

**Remark 1** (Extensions). *While Client-Selection is defined over a box constraint $\boldsymbol{W} \in [0, 1]^{n \times n}$, it can be easily extended to other convex domains. For example, in all-for-one type collaborative training, the weights are optimized over a simplex. The experiment on language models is deferred to Section 4.3.*

## 2.1 Algorithm

We propose a novel stochastic gradient descent (SGD)-type alternating optimization algorithm, termed CoBo, to solve the bilevel optimization problem defined by (Model-Training) and (Client-Selection). The algorithm alternates between updating the model variables $\boldsymbol{X}$ and the collaboration weights $\boldsymbol{W}$.

In each iteration $t$, we first fix the model variables $\{\boldsymbol{x}_i^t\}_{i=1}^n$ and update the collaboration weights by applying projected gradient ascent with step size $\gamma > 0$ to the inner problem (Client-Selection):

$$w_{ij}^{t+1} = \text{Proj}_{[0,1]}\left(w_{ij}^t + \gamma \left\langle \nabla f_i\left(\frac{\boldsymbol{x}_i^t + \boldsymbol{x}_j^t}{2}\right), \nabla f_j\left(\frac{\boldsymbol{x}_i^t + \boldsymbol{x}_j^t}{2}\right) \right\rangle\right) \qquad \forall i, j \in [n]. \quad (1)$$

Next, with the updated collaboration weights $\{w_{ij}^{t+1}\}$ are fixed, we optimize the model variables $\{\boldsymbol{x}_i\}_{i=1}^n$ using the following update rule derived from the outer problem (Model-Training):

$$\boldsymbol{x}_i^{t+1} = \boldsymbol{x}_i^t - \eta\left(\nabla f_i(\boldsymbol{x}_i^t) + \rho \sum_{k=1}^n w_{ik}^{t+1}\left(\boldsymbol{x}_i^t - \boldsymbol{x}_k^t\right)\right) \qquad \forall i \in [n], \quad (2)$$

where $\eta > 0$ is the step size for updating the model variables. This alternating process is repeated until convergence.

The detailed implementation of the algorithm is provided in Algorithm 1. In this implementation, the full gradients $\{\nabla f_i\}_{i \in [n]}$ in (1) and (2) are replaced by their stochastic estimates. Additionally, collaborative weights are updated with a probability of $\mathcal{O}(\frac{1}{n})$, leading to an expected computation of

---

**Algorithm 1** COBO: **Co**llaborative Learning via **B**ilevel **O**ptimization

---

**Input:** Model parameters $\forall i \in [n] \; \boldsymbol{x}_i^0 = \boldsymbol{x}^0 \in \mathbb{R}^d$; Penalization parameter $\rho > 0$; $\boldsymbol{W}^0 \in \mathbb{R}^{n \times n}$ where $w_{ij}^0 = 1, \forall i, j \in [n]$; Step size $\eta, \gamma > 0$.

 1: **for** round $t = 0, 1 \ldots, T$ **do**
 2:     Call $\boldsymbol{W}^{t+1} \leftarrow$ Client-Selection$(\{\boldsymbol{x}_i^t\}_{i \in [n]}, \boldsymbol{W}^t)$
 3:     **for** client $i = 1, \ldots n$ **do**
 4:         Draw sample $\xi_i \sim \mathcal{D}_i$ and compute stochastic gradient $\boldsymbol{g}_i^t \in \mathbb{R}^d$ of $f_i(\boldsymbol{x}_i^t)$ and update

$$\boldsymbol{x}_i^{t+1} \leftarrow \boldsymbol{x}_i^t - \eta \left( \boldsymbol{g}_i^t + \rho \sum_{k=1}^n w_{ik}^{t+1} \left( \boldsymbol{x}_i^t - \boldsymbol{x}_k^t \right) \right) \tag{3}$$

 5:     **end for**
 6: **end for**
 7: **Output:** Uniform randomly select $s \in [T]$ and return $\{\boldsymbol{x}_0^s, \ldots, \boldsymbol{x}_n^s\}$ and $\boldsymbol{W}^s$.
 8:
 9: **procedure** CLIENT-SELECTION$(\boldsymbol{X}, \boldsymbol{W})$
10:     **for** each pair of clients $(i, j)$ where $i \neq j \in [n]$ **do**
11:         **if** with a probability $1/n$, **then**
12:             Compute the average model $\boldsymbol{z}_{ij} = \frac{1}{2}(\boldsymbol{x}_i + \boldsymbol{x}_j)$.
13:             Compute stochastic gradient $\boldsymbol{g}_{i \leftarrow i}$ and $\boldsymbol{g}_{i \leftarrow j}$ for $f_i(\boldsymbol{z}_{ij})$ and $f_j(\boldsymbol{z}_{ij})$ respectively,

$$w_{ij} \leftarrow \text{Proj}_{[0,1]} \left( w_{ij} + \gamma \left\langle \boldsymbol{g}_{i \leftarrow i}, \boldsymbol{g}_{i \leftarrow j} \right\rangle \right). \tag{4}$$

14:         **end if**
15:     **end for**
16:     **return** updated selection variables $\boldsymbol{W}$
17: **end procedure**

---

$\mathcal{O}(n)$ gradients. This results in an overhead comparable to that of standard decentralized learning methods [25, 19], thereby enabling client selection with minimal additional cost.

Compared to federated clustering algorithms, which require global synchronization before applying clustering oracles, the inner problem (Client-Selection) in COBO is solved in a pairwise fashion. This pairwise approach makes the algorithm non-blocking and robust to stragglers, providing greater flexibility and efficiency. Not all pairwise weights have to be computed in each iteration. In Table 1 we compare the performance of multiple edge-sampling strategies.

## 3    Theoretical results

In this section, we define assumptions in collaborative learning settings and show that COBO converges to stationary points. The following assumptions regarding the local optimization objective $f_i$ are commonly adopted in the literature [1, 19]:

**(A1) $L$-smooth.** *For all $\boldsymbol{x}$ and $\boldsymbol{y}$ in $\mathbb{R}^d$ and $i \in [n]$, the loss function $f_i$ has $L$-Lipschitz gradients, i.e.*

$$\|\nabla f_i(\boldsymbol{x}) - \nabla f_i(\boldsymbol{y})\| \leq L \|\boldsymbol{x} - \boldsymbol{y}\| .$$

**(A2) Noise bound.** *For all $\boldsymbol{x} \in \mathbb{R}^d$ and $i \in [n]$, there exists $\sigma^2 > 0$ such that the stochastic gradient has bounded noise*

$$\mathbb{E}_\xi \left[ \|\nabla f_i(\boldsymbol{x}; \xi) - \mathbb{E}_\xi \left[ \nabla f_i(\boldsymbol{x}; \xi) \right]\|^2 \right] \leq \sigma^2 .$$

**(A3) Global minimum.** *For all $i \in [n]$, the loss function $f_i$ has a global lower bound $f_i^*$.*

The next assumption characterizes the possible relationships between clients. In the first case, when reaching the stationary point $\boldsymbol{x}$ of their joint objective $f_i + f_j$, then by (5) implies that $\nabla f_i(\boldsymbol{x}) = \nabla f_j(\boldsymbol{x}) = \boldsymbol{0}$ client $i$ and $j$ reach their own stationary points. In the second case, when client $i$ reaches its stationary point, the gradient of $j$ is lower bounded by a positive constant, meaning they don't share stationary points. This leads to eventual

**(A4) Collaborativeness.** *If clients $i$ and $j$ are collaborative, then there exists $M_{ij} > 0$ such that*

$$\|\nabla f_i(\boldsymbol{x}) - \nabla f_j(\boldsymbol{x})\|_2^2 \le M_{ij}^2 \|\nabla f_i(\boldsymbol{x}) + \nabla f_j(\boldsymbol{x})\|_2^2 \qquad \forall\, \boldsymbol{x} \in \mathbb{R}^d. \tag{5}$$

*Otherwise, there exists $\zeta_{ij}^2 > 0$ such that*

$$\|\nabla f_i(\boldsymbol{x})\|_2^2 + \|\nabla f_j(\boldsymbol{x})\|_2^2 \ge \zeta_{ij}^2 \qquad \forall\, \boldsymbol{x} \in \mathbb{R}^d. \tag{6}$$

This assumption is similar to [39, Assumptions 4,5], but we define relations for pairs of clients instead of clusters. In the next example, we use quadratics to demonstrate (A4)

**Example 2.** *Assume that there are $K$ clusters with $[n] = \cup_{k \in [K]} \mathcal{C}_k$ and $\mathcal{C}_k \cap \mathcal{C}_{k'} = \emptyset$ for all $k \ne k' \in [K]$. Consider the $k$-th cluster with center $\boldsymbol{\mu}_k$ and client $i \in \mathcal{C}_k$, the loss function is $f_i(\boldsymbol{x}) = \frac{a_i}{2}\|\boldsymbol{x} - \boldsymbol{\mu}_k\|_2^2$ where $a_i > 0$. Then for clients $i, j$ in the same cluster, i.e. $i, j \in \mathcal{C}_k$*

$$\|\nabla f_i(\boldsymbol{x}) - \nabla f_j(\boldsymbol{x})\|_2^2 = (a_i - a_j)^2 \|\boldsymbol{x} - \boldsymbol{\mu}_k\|_2^2 = \frac{(a_i - a_j)^2}{(a_i + a_j)^2} \|\nabla f_i(\boldsymbol{x}) + \nabla f_j(\boldsymbol{x})\|_2^2.$$

*The $M_{ij} = \frac{|a_i - a_j|}{a_i + a_j}$ in this case. On the other hand, for $i \in \mathcal{C}_k$ and $j \in \mathcal{C}_{k'}$ and $\boldsymbol{\mu}_k \ne \boldsymbol{\mu}_{k'}$,*

$$\|\nabla f_i(\boldsymbol{x})\|_2^2 + \|\nabla f_j(\boldsymbol{x})\|_2^2 = a_i^2 \|\boldsymbol{x} - \boldsymbol{\mu}_k\|_2^2 + a_j^2 \|\boldsymbol{x} - \boldsymbol{\mu}_{k'}\|_2^2 = \frac{a_i^2 a_j^2}{(a_i^2 + a_j^2)^2} \|\boldsymbol{\mu}_k - \boldsymbol{\mu}_{k'}\|_2^2$$

*where the lower bound $\zeta_{ij}^2 = \frac{a_i^2 a_j^2}{(a_i^2 + a_j^2)^2} \|\boldsymbol{\mu}_k - \boldsymbol{\mu}_{k'}\|_2^2 > 0$.*

Finally, we derive a convergence theorem with the assumption that clients are drawn from clusters, as e.g. in [33, Assumption 2].

**(A5) Cluster.** *All clients are drawn from clusters where within each cluster clients share stationary points.*

**Theorem I.** *Suppose Assumption 1,2,3,4,5 hold true. Suppose that CoBo solves (4) with mini-batch size $b$. Consider clients $i$ and $j$ in the same cluster $\mathcal{C}$ of size $c$. Suppose that $M_{ij}^2 \in (0, \frac{1}{5})$, $b \ge \frac{2}{c^2} 2L\eta(c-2)\sigma^2$ and $\zeta_{ik}^2 \ge \|\nabla f_i(\boldsymbol{x}) + \nabla f_k(\boldsymbol{x})\|_2^2$ for all $\boldsymbol{x}$ and $k$. Let $\rho \ge \frac{\sqrt{3}L}{c}$ and step size*

$$\eta \le \min\left\{ \frac{2}{\sigma\sqrt{LT}} \sqrt{\frac{1}{c^2} \sum_{i,j \in \mathcal{C}} \left(\tilde{f}_{ij}\left(\boldsymbol{z}_{ij}^0\right) - \tilde{f}_{ij}^\star\right)},\, \frac{1}{2\sqrt{3}L} \right\}.$$

*The consensus distance also converges to 0, i.e.*

$$\frac{1}{c^2 T} \sum_{t=0}^{T-1} \sum_{i,j \in \mathcal{C}} \mathbb{E}\left[\|\boldsymbol{x}_i^{t+1} - \boldsymbol{x}_j^{t+1}\|_2^2\right] \le \frac{6M_{ij}^2}{\rho^2 c^2} \sqrt{\frac{L\sigma^2}{c^2 T} \sum_{i,j \in \mathcal{C}} \left(\tilde{f}_{ij}\left(\boldsymbol{z}_{ij}^0\right) - \tilde{f}_{ij}^\star\right)}.$$

*Moreover, the gradient norm is upper bounded.*

$$\frac{1}{c^2 T} \sum_{t=0}^{T-1} \sum_{i,j \in \mathcal{C}} \mathbb{E}\left[\left\|\nabla \tilde{f}_{ij}\left(\boldsymbol{z}_{ij}^t\right)\right\|_2^2\right] \le 3 \sqrt{\frac{L\sigma^2}{c^2 T} \sum_{i,j \in \mathcal{C}} \left(\tilde{f}_{ij}\left(\boldsymbol{z}_{ij}^0\right) - \tilde{f}_{ij}^\star\right)}.$$

This theorem suggests that clients inside the same cluster gradually reach consensus. This cluster-level consensus model reaches stationary point of their losses by (5). Note that a larger penalization parameter $\rho$ and smaller values of $M_{ij}^2$ lead to faster convergence, which aligns with our expectations. Note that $M_{ij}$ in (A4) measures how well i,j collaborate. A smaller $M_{ij}$ leads to better consensus distance in Theorem I, with $M_{ij} = 0$ leading to identical data distribution. The following corollary states the convergence of norm of client gradient of model $\boldsymbol{x}_i$.

**Corollary II.** *Under same conditions as Theorem I, $\|\nabla f_i\left(\boldsymbol{x}_i^t\right)\|_2^2$ converges at a similar rate*

$$\frac{1}{c^2 T} \sum_{t=0}^{T-1} \sum_{i,j \in \mathcal{C}} \|\nabla f_i\left(\boldsymbol{x}_i^t\right)\|_2^2 \le 4 \sqrt{\frac{L\sigma^2}{c^2 T} \sum_{i,j \in \mathcal{C}} \left(\tilde{f}_{ij}\left(\boldsymbol{z}_{ij}^0\right) - \tilde{f}_{ij}^\star\right)}.$$

# 4   Experiments

In this section, we present three experiments to demonstrate the practical effectiveness of COBO. In the first two experiments, we benchmark COBO in both a cross-silo federated learning setup involving 8 clients and a cross-device setup with 80 clients, using the CIFAR-100 dataset for multi-task learning [21]. In the third experiment, we train language models on subsets of the Wiki-40B dataset while learning domain weights within a simplex [12]. Compared to state-of-the-art personalized federated learning baselines, COBO obtains personalized models of higher quality and correctly identifies cluster structures. Details of the experiments, including descriptions of the architectures and the system setup, are deferred to Appendix B.

Throughout the experiments, we use popular federated learning baselines such as FEDAVG [28], Federated Clustering (abbreviated as FC) [39], DITTO [24], IFCA [11], and an oracle algorithm. The definition of the oracle baseline varies in each setup and will be discussed case by case. Note that we additionally provide clustering-based algorithms, i.e., FC and IFCA, with the actual number of clusters. Their experimental statistics reported in this section, such as accuracy and perplexity, include this advantage. In addition to previous baselines, we also compare COBO with a collaborative fine-tuning approach for large language models that leverages performance on validation data to determine the collaboration weights  [38] (referred to as Validation Based in Table 2).

## 4.1   Cross-silo federated learning experiment with 8 clients

In this experiment, we evaluate the performance of COBO by comparing the average accuracies of local models against those of established collaborative learning baselines. Our objective is to assess how effectively COBO discerns and leverages the structure of data clusters relative to other collaborative learning algorithms.

We simulate a cross-silo multi-task environment where training a single model across all clients yields poor performance, thus highlighting the necessity for client selection. Our experimental configuration consists of 4 clusters, each containing 2 clients utilizing the ResNet-9 model [13]. To encourage collaboration within clusters, we randomly allocate half of the dataset to each client in a cluster. To differentiate between clusters, we introduce label diversity by flipping the image labels in each cluster using distinct random seeds. This process ensures that each class maintains unique labels across all clusters, effectively creating a scenario where a universally trained model would not be optimal, thereby necessitating personalized models that can cater to the specific label distribution of each cluster.

In this context, collaboration among clients within the same cluster is advantageous, as their datasets are complementary. There are two primary reasons why collaboration between different clusters may not be beneficial: (1) the dataset available to clients within each cluster is identical, negating the incentive to collaborate with clients from other clusters; and (2) the label flipping across clusters could mean that inter-cluster collaboration might actually degrade local model performance.

Given these considerations, we designate an oracle algorithm for our scenario: FEDAVG implemented separately within each cluster. This ensures that collaboration is confined to where it is most beneficial. Additionally, the oracle collaboration matrix is defined to be a block-diagonal matrix, with entries of 1 for pairs of clients within the same cluster (indicating collaboration) and entries of 0 for pairs from different clusters (indicating no collaboration). This matrix serves as a benchmark for the ideal collaboration structure in our simulated environment.

To enable the practical application of COBO, we sample pairs of clients in each iteration to update their collaboration weights. We begin by examining the impact of various sampling strategies on the performance of COBO. The primary approach involves sampling with a constant probability of $\mathcal{O}(1/n)$. Additionally, we observe that COBO identifies an appropriate collaboration matrix early in the training process, motivating the use of a time-step-dependent sampling rate, $\mathcal{O}(1/t)$. We also implement a mixed strategy: employing the constant sampling rate, $\mathcal{O}(1/n)$, for the initial 0.2% of iterations, followed by a switch to the time-dependent sampling rate, $\mathcal{O}(1/t)$, for the remainder of the training. A comparison of these strategies with the non-sampling oracle, where all pairs are updated in every iteration, is presented in Table 1. While COBO demonstrates consistent performance across all sampling strategies, achieving results close to those of the non-sampling oracle, the mixed strategy shows a slight performance advantage.

Table 1: Comparison of the average performance of CoBo across different sampling strategies for updating the weights of client pairs in the collaboration matrix. All strategies demonstrate performance close to that of the non-sampling oracle. However, the mixed strategy, which combines a constant sampling rate at the start with a time-dependent rate during later training phases, shows superior performance.

|  | Acc.(%) | Loss |
|---|---|---|
| Constant ($\mathcal{O}(1/n)$) | 73.05 | 1.104 |
| Time-dependent ($\mathcal{O}(1/t)$) | 73.18 | 1.226 |
| Mixed | 74.77 | 1.081 |
| No Sampling (Oracle) | 74.93 | 1.278 |

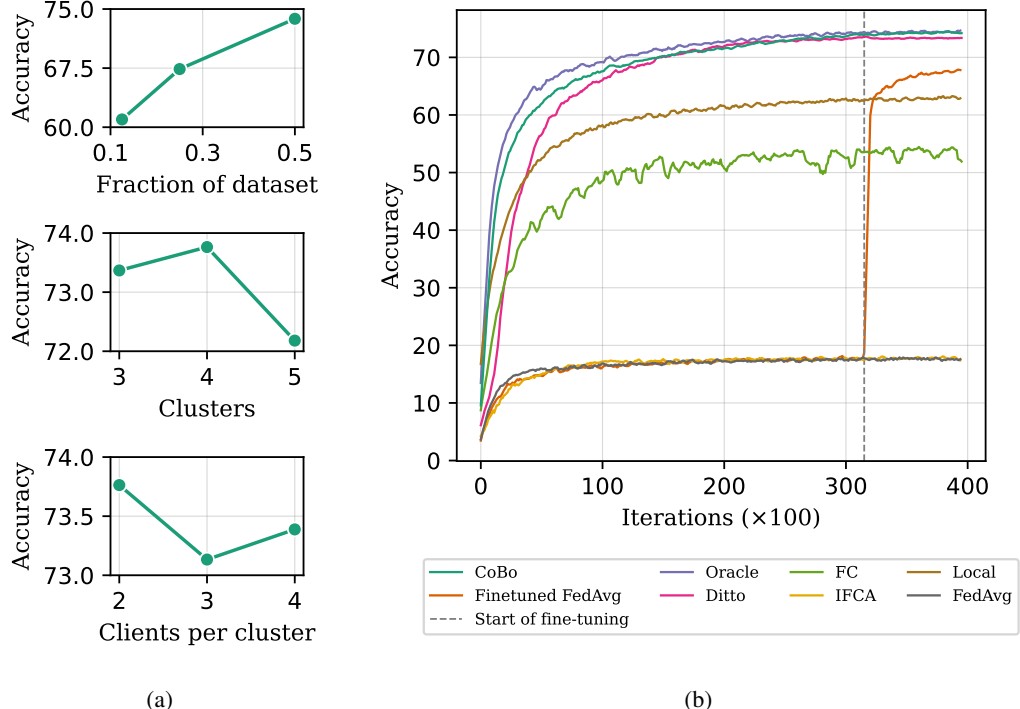

(a)                                          (b)

Figure 2: (2a) Average accuracy in cross-silo experiments with varying factors, including the fraction of the dataset available to clients, the number of clusters, and the number of clients per cluster. (2b) Average accuracy of personalized models for cross-silo federated learning with 8 clients. The "Oracle" denotes applying FedAvg to the clients with the same label permutation.

To further assess performance, we trained CoBo and other baseline algorithms for a total of 40,000 iterations. Figure 2b presents the accuracy diagram. We observe that CoBo almost reaches the performance bound established by the Oracle. Moreover, CoBo achieves a fixed accuracy of 60% in 4,500 iterations, which is 30% faster than Ditto. For better comparison, the values of accuracy and loss are reported in Table 2. Additionally, the evolution of the collaboration matrix for clustering algorithms and CoBo is illustrated in Figure 3. CoBo starts to identify clients with similar label permutations as early as 300 iterations and stabilizes in less than 5,000 iterations (12.5% of the training phase). IFCA always degenerates to one fully connected cluster, while FC periodically suffers from clustering mistakes even at the end of training.

Figure 2a presents the results of the cross-silo experiment under various configurations to further assess the robustness of CoBo. First, we modify the fraction of the dataset allocated to each client. Intuitively, the total amount of data available to a cluster directly impacts the performance of CoBo. Next, we experiment with different numbers of clusters, each containing two clients, and observe that

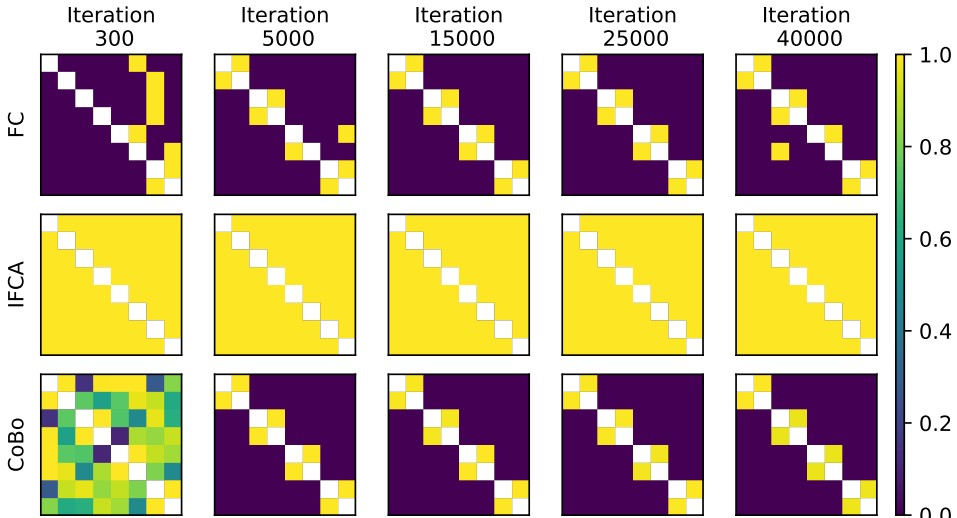

Figure 3: Collaboration matrices learned by Federated Clustering (FC), IFCA, and CoBo at different stages of training for cross-silo experiment with 8 clients. The diagonals are masked out. The oracle matrix is a block diagonal matrix with blocks of size 2. The collaboration matrix of CoBo already starts to look similar to oracle matrix within as low as 300 iterations (0.75% of the total iterations), and converges to it within 5000 iterations (12.5% of the total iterations). On the other hand, IFCA yields a fully-connected matrix while FC occasionally diverges from the achieved cluster structures (e.g., iterations 300, 5000, and 40000), even at the end of training.

the number of clusters does not significantly affect CoBo 's accuracy. Additionally, we investigate the effect of varying the number of clients per cluster while maintaining a fixed total of four clusters. In this setup, the dataset is partitioned among clients within each cluster, resulting in less data per client as the cluster size increases. Despite this, CoBo leverages collaboration to maintain robust performance even with larger cluster sizes.

## 4.2 Cross-device experiment experiment with 80 clients

In this experiment, we demonstrate the performance of CoBo in a challenging cross-device federated learning setting characterized by significant data heterogeneity. We create 10 clusters of varying sizes: 2 clusters consist of 6 clients each, another 2 comprise 7 clients each, and so on. Each cluster is allocated data from 10 distinct classes out of the total 100 classes available in the CIFAR-100 dataset, ensuring that the data across clusters are disjoint. Within each cluster, the data are distributed uniformly at random among the clients. We proceed to train individual ResNet-9 models [13] on each client's data for a total of 20,000 iterations. This setup allows us to observe the behavior of CoBo and its ability to handle both the quantity and diversity of data across different client groups and cluster sizes.

We define the oracle algorithm and the corresponding collaboration matrix in the same manner as in Section 4.1. Note that while we manually create the clusters, inter-cluster collaboration may still be helpful in practice, and it is impossible to know the actual ground truth in this case. Consequently, we recognize that the oracle may not correspond to the optimal performance. Nevertheless, this oracle still exhibits superior performance compared to other baselines that lack prior knowledge of the data distribution among clients, as evidenced by the results presented in Table 2. The collaboration matrix and accuracy plots are deferred to Figure 5 and Figure 6 in Appendix B, respectively.

In this challenging experiment, CoBo surpasses all other baselines by at least 5.7% in accuracy. This supports the conclusion that CoBo scales well with the size of collaborative learning and effectively exploits collaboration weights among clients at a fine-grained level.

Table 2: Comparisons of model quality and fairness measure of personalized models for cross-silo experiment with 8 clients, and cross-device experiment with 80 clients, and the language modelling experiment with 4 clients having different languages. Federated clustering (FC) is not scalable with number of clients due to its $\mathcal{O}(n^2)$ complexity, and therefore ignored in the cross-device fl experiment. The clustering algorithms IFCA and FC are not applicable to LLMs and there ignored. Note that Oracle is not defined in the LLMs experiment. The column "Imp.(%)" demonstrates the percentage of clients with improved performance compared to local training.

|  | Cross-silo | | | Cross-device | | | Fine-tuning LLMs | |
|---|---|---|---|---|---|---|---|---|
|  | Acc.(%) | Loss | Imp.(%) | Acc.(%) | Loss | Imp.(%) | Perplexity | Imp.(%) |
| Local | 64.9 ± 0.1 | 1.67 | - | 54.9 ± 0.1 | 1.40 | - | 41.26 ± 0.38 | - |
| FedAvg | 18.8 ± 0.1 | 2.66 | 0 | 53.9 ± 0.1 | 1.79 | 29 | 64.84 ± 0.00 | 0 |
| Fine-tuning FedAvg | 70.2 ± 0.2 | 1.77 | 0 | 58.9 ± 0.1 | 1.88 | 94 | 46.70 ± 0.07 | 0 |
| Ditto | 73.5 ± 0.3 | 1.55 | **100** | 70.3 ± 0.1 | 1.21 | **100** | 40.05 ± 0.01 | **100** |
| IFCA | 18.6 ± 0.1 | 2.75 | 0 | 45.6 ± 0.8 | 2.15 | 4 | - | - |
| FC | 55.1 ± 0.4 | 1.79 | 0 | - | - | - | - | - |
| Validation Based | - | - | - | - | - | - | 42.90 ± 1.68 | 75 |
| CoBo | **74.6 ± 0.2** | **1.08** | **100** | **79.6 ± 0.4** | **0.97** | **100** | **39.28 ± 0.01** | **100** |
| Oracle | 75.4 ± 0.2 | 1.07 | 100 | 83.6 ± 0.3 | 0.70 | 100 | - | - |

## 4.3 Collaborative fine-tuning on language models

Recently, Large Language Models (LLMs) have gained significant popularity due to their ability to effectively solve challenging tasks. Their downstream performance can be further enhanced by fine-tuning; however, the scarcity of data often leads to inferior performance and necessitates collaboration. Therefore, we conduct an experiment with four clients, each having a pre-trained GPT-2 base model[3] with 124 million parameters in total [32], and a subset of articles from the Wiki-40B dataset [12] in one of the following four languages: Catalan, Spanish, German, or Dutch. We use LoRA for the Self-Attention and MLP layers for fine-tuning, which accounts for 0.47% of the full parameters [14].

For data-hungry tasks, such as those involving LLMs, contributions from all domains are valuable. Clustering methods fall short in this aspect due to their binary, discrete outputs, which do not capture the nuanced degrees of collaboration needed. CoBo addresses this limitation by allowing for a continuous range of collaboration intensities, achieved by a simple yet effective modification to the projection domain in (1). Specifically, we employ a probability simplex, denoted as $\Delta_i = \{w_{ij} \geq 0, \sum_j w_{ij} = 1\}$, as the domain of the inner problem.

In Table 2, we compare the perplexity of CoBo with baselines after 500 iterations, when FEDAVG converges. There are no oracle domain weights in this experiment due to the complex coherence among languages; therefore, we omit the oracle algorithm in the table. CoBo achieves the best perplexity among all algorithms. In Figure 4, we demonstrate the domain weights learned for the Catalan language. Overall, Catalan assigns the highest collaboration weight to Spanish, which is reasonable considering the similarity between the two languages.

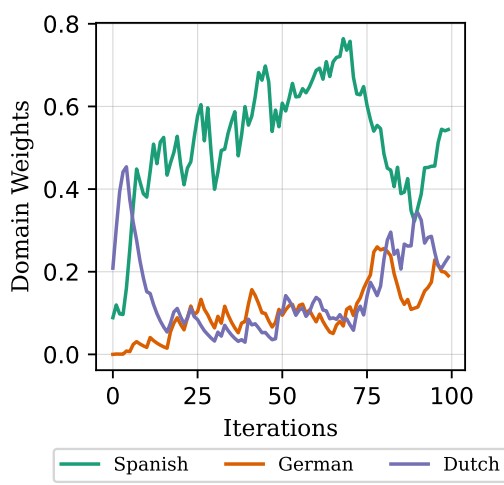

Figure 4: Domain weights found by CoBo for Catalan language. There are 4 domains in total: Catalan, Spanish, German, and Dutch. The curves are smoothed by exponential moving average.

---

[3] https://github.com/karpathy/nanoGPT

# 5 Related Work

**Personalized federated learning.** Personalized federated learning has received significant attention in recent years due to its potential to tailor models to individual user data while benefit from collaboration [33, 35, 36, 9, 22, 3]. There are various flavors of personalized federated learning. DITTO trains personalized models by incorporating a regularization term that penalizes the divergence from a global model [24]. Many personalization works assume that clients are drawn from clusters. For example, Marfoq et al. [27] use K-nearest neighbors (KNN) to determine collaborators. Mansour et al. [26], Ghosh et al. [11], Werner et al. [39] develop $K$ personalized models and assign clients to clusters based on criteria such as minimum function values or gradient similarities. Additionally, Even et al. [7] provided theoretical insights by establishing lower bounds, which demonstrate that the optimal gradient filtering strategy involves clustering clients with identical optima.

**Federated learning with client selection** In federated learning, client selection is often performed by simultaneously minimizing task losses and collaborative weights in a single-level objective function. Zantedeschi et al. [40] minimize task losses augmented with a penalization term $w_{ij}\|\boldsymbol{x}_i - \boldsymbol{x}_j\|_2^2$, similar to our outer problem. However, optimizing $w_{ij}$ directly can lead to a degenerate solution ($w_{ij} = 0$), which necessitates an additional penalization for small $w_{ij}$ values. Smith et al. [34] approach multi-task learning by minimizing task losses with a more sophisticated penalization term that accounts for the relationships between tasks. This formulation requires the client-selection function to be consistent with client selection, which can negatively impact performance. Apart from multi-task federated learning, a similar bilevel optimization formulation has been used by Le Bars et al. [23] to find a sparse mixing matrix while training a consensus model in the outer problem.

**Bilevel optimization and alternating optimization.** Bilevel optimization is a powerful tool which models a broad range of problems, such as reinforcement learning [6, 30, 16, 15, 37, 31], and linearly-solvable Markov decision process [5], meta-learning [8, 20], etc. A typical bilevel optimization problem, as the name indicates, consists of an outer and an inner optimization problem whose variables are inter-dependent. Typical bilevel optimization solvers requires hessian information which is usually expansive to acquire [8]. On the other hand, alternating optimization tools has been used be used to solve bilevel optimization problem [2, 4]. While in general there is no universal convergence guarantees for alternative optimizations, the special structure of our inner problem ensures the convergence of COBO to the stationary point.

# 6 Conclusions

Existing collaborative learning algorithms only allow coarse-grained collaboration, which leads to inferior performance in practice. To address this issue, we model collaborative learning as a special bilevel optimization problem where client selection is based on the optimization of a linear function of gradient alignment measure for each pair of clients. In addition, we propose an efficient SGD-type alternating optimization algorithm COBO which is scalable, elastic, and enjoy theoretical guarantees. Besides, COBO empirically outperforms popular personalized federated learning algorithms in realistic collaborative learning problems.

**Limitations.** In this work, we do not take privacy into consideration. The existing algorithm requires exchanging gradients between collaborators when updating weight $\boldsymbol{w}$ which may raise privacy concerns. We defer the discussion of privacy-preserving collaborative learning framework to future work.

**Acknowledgement.** We acknowledge funding from Swiss National Science Foundation (SNSF) grant number 200020_200342, from Huawei Cloud Intelligent Cloud Technologies Initiative, and from Google Research Collaborations.

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

# A Theory

Let $z_{ij}^t := \frac{1}{2}(x_i^t + x_j^t)$ be the average iterate of $x_i^t$ and $x_j^t$ and $\tilde{f}_{ij}(x) := \frac{1}{2}(f_i(x) + f_j(x))$ be an averaged objective function.

**Lemma 3.** *Suppose (A1) holds true. Let $\eta \leq \frac{1}{2L}$. Then for $i, j$ in the same cluster $\mathcal{C}$ of size $c$*

$$\frac{1}{c^2} \sum_{i,j \in \mathcal{C}} \left\| \nabla \tilde{f}_{ij}\left(z_{ij}^t\right) \right\|_2^2 \leq \frac{2}{\eta} \frac{1}{c^2} \sum_{i,j \in \mathcal{C}} \left( \tilde{f}_{ij}\left(z_{ij}^t\right) - \mathbb{E}[\tilde{f}_{ij}\left(z_{ij}^{t+1}\right)] \right) + D_1 \frac{1}{c^2} \sum_{i,j \in \mathcal{C}} \left\| x_i^t - x_j^t \right\|_2^2$$

$$+ 3n^2 \rho^2 \frac{1}{cn} \sum_{i \in \mathcal{C}} \sum_{k=1}^{n} |\mathbb{E}_h[w_{ik}^{t+1}] - w_{ik}^\star|^2 \left\| x_i^t - x_k^t \right\|_2^2 + \frac{L\eta\sigma^2}{2}.$$

*where coefficient $D_1$ is defined as*

$$D_1(L, c, \eta, b, \sigma^2) := \frac{3L^2}{4} + 3c^2\rho^2 + \frac{L\eta\rho^2(c-2)2\sigma^2}{b}.$$

## A.1 Proof of Lemma 3

*Proof.* Let $h_i^t$ and $h_j^t$ be independent and unbiased estimates of $\nabla f_i(z_{ij}^t)$, $\nabla f_j(z_{ij}^t)$ respectively. The variance of $h_i^t$ has a variance of $\frac{\sigma^2}{b}$. Let's denote $\mathbb{E}_g[\cdot] := \mathbb{E}_{g_1,\ldots,g_n}[\cdot | z_i^t]$ and $\mathbb{E}_h[\cdot] := \mathbb{E}_{h_1,\ldots,h_n}[\cdot]$ and let $\mathbb{E}[\cdot] = \mathbb{E}_h[\mathbb{E}_g[\cdot]]$. By the L-smoothness assumption (A1) and bounded noise assumption (A2)

$$\mathbb{E}_h \mathbb{E}_g \left[ \tilde{f}_{ij}\left(z_{ij}^{t+1}\right) \right] \leq \tilde{f}_{ij}\left(z_{ij}^t\right) + \left\langle \nabla \tilde{f}_{ij}\left(z_{ij}^t\right), \mathbb{E}_h \mathbb{E}_g \left[ z_{ij}^{t+1} - z_{ij}^t \right] \right\rangle + \frac{L}{2} \|\mathbb{E}_h \mathbb{E}_g[z_{ij}^{t+1} - z_{ij}^t]\|_2^2$$

$$+ \frac{L}{2} \mathbb{E}_h \mathbb{E}_g \left[ \|z_{ij}^{t+1} - z_{ij}^t - \mathbb{E}_h \mathbb{E}_g[z_{ij}^{t+1} - z_{ij}^t]\|_2^2 \right].$$

The last quantity, i.e. variance of $z_{ij}^{t+1} - z_{ij}^t$, can be expanded as follows

$$\mathbb{E}_h \mathbb{E}_g \left[ \|z_{ij}^{t+1} - z_{ij}^t - \mathbb{E}_h \mathbb{E}_g[z_{ij}^{t+1} - z_{ij}^t]\|_2^2 \right] = \mathbb{E}_h \mathbb{E}_g \left[ \|z_{ij}^{t+1} - \mathbb{E}_h \mathbb{E}_g[z_{ij}^{t+1}]\|_2^2 \right]$$

$$= \mathbb{E}_h \mathbb{E}_g \left[ \|z_{ij}^{t+1} \pm \mathbb{E}_h[z_{ij}^{t+1}] - \mathbb{E}_h \mathbb{E}_g[z_{ij}^{t+1}]\|_2^2 \right]$$

$$= \mathbb{E}_g \left[ \|\mathbb{E}_h[z_{ij}^{t+1}] - \mathbb{E}_h \mathbb{E}_g[z_{ij}^{t+1}]\|_2^2 \right] + \mathbb{E}_h \mathbb{E}_g \left[ \|z_{ij}^{t+1} - \mathbb{E}_h[z_{ij}^{t+1}]\|_2^2 \right]$$

$$= \mathbb{E}_g \left[ \|\mathbb{E}_h[z_{ij}^{t+1}] - \mathbb{E}_h \mathbb{E}_g[z_{ij}^{t+1}]\|_2^2 \right] + \mathbb{E}_h \mathbb{E}_g \left[ \|z_{ij}^{t+1} - \mathbb{E}_h[z_{ij}^{t+1}]\|_2^2 \right]$$

$$= \eta^2 \mathbb{E}_g \left\| \frac{1}{2}(g_i^t + g_j^t - \nabla f_i(x_i^t) - \nabla f_j(x_j^t)) \right\|_2^2 + \mathbb{E}_h \mathbb{E}_g \left[ \|z_{ij}^{t+1} - \mathbb{E}_h[z_{ij}^{t+1}]\|_2^2 \right]$$

$$\leq \frac{\eta^2 \sigma^2}{4} + \underbrace{\mathbb{E}_h \mathbb{E}_g \left[ \|z_{ij}^{t+1} - \mathbb{E}_h[z_{ij}^{t+1}]\|_2^2 \right]}_{=:\mathcal{T}_{ij}}$$

where the linear term vanishes due to the expectation of $h$ in the second equality and the definition of $z_{ij}^{t+1}$ and (3) for the third equality. Then

$$\mathbb{E} \left[ \tilde{f}_{ij}\left(z_{ij}^{t+1}\right) \right] \leq \tilde{f}_{ij}\left(z_{ij}^t\right) + \left\langle \nabla \tilde{f}_{ij}\left(z_{ij}^t\right), \mathbb{E}\left[z_{ij}^{t+1} - z_{ij}^t\right] \right\rangle + \frac{L}{2} \|\mathbb{E}[z_{ij}^{t+1} - z_{ij}^t]\|_2^2 + \frac{L\eta^2\sigma^2}{4} + \frac{L}{2}\mathcal{T}_{ij}.$$

Expand the inner product with equality $-\langle x, y \rangle = -\frac{1}{2}\|x\|_2^2 - \frac{1}{2}\|y\|_2^2 + \frac{1}{2}\|x - y\|_2^2$ yields

$$\mathbb{E} \left[ \tilde{f}_{ij}\left(z_{ij}^{t+1}\right) \right] \leq \tilde{f}_{ij}\left(z_{ij}^t\right) + \frac{\eta}{2} \left\| \frac{\mathbb{E}[z_{ij}^{t+1}] - z_{ij}^t}{\eta} + \nabla \tilde{f}_{ij}\left(z_{ij}^t\right) \right\|_2^2 - \frac{\eta}{2} \left\| \frac{\mathbb{E}[z_{ij}^{t+1}] - z_{ij}^t}{\eta} \right\|_2^2$$

$$- \frac{\eta}{2} \left\| \nabla \tilde{f}_{ij}\left(z_{ij}^t\right) \right\|_2^2 + \frac{L\eta^2}{2} \left\| \frac{\mathbb{E}[z_{ij}^{t+1}] - z_{ij}^t}{\eta} \right\|_2^2 + \frac{L\eta^2\sigma^2}{4} + \frac{L}{2}\mathcal{T}_{ij}$$

$$\leq \tilde{f}_{ij}\left(z_{ij}^t\right) + \frac{\eta}{2} \left\| \frac{\mathbb{E}[z_{ij}^{t+1}] - z_{ij}^t}{\eta} + \nabla \tilde{f}_{ij}\left(z_{ij}^t\right) \right\|_2^2$$

$$- \frac{\eta}{4} \left\| \frac{\mathbb{E}[z_{ij}^{t+1}] - z_{ij}^t}{\eta} \right\|_2^2 - \frac{\eta}{2} \left\| \nabla \tilde{f}_{ij}\left(z_{ij}^t\right) \right\|_2^2 + \frac{L\eta^2\sigma^2}{4} + \frac{L}{2}\mathcal{T}_{ij}$$

where $\eta \leq \frac{1}{2L}$ is applied in the last inequality. Then the upper bound of $\left\|\nabla \tilde{f}_{ij}\left(\boldsymbol{z}_{ij}^{t}\right)\right\|_{2}^{2}$ is that

$$
\left\|\nabla \tilde{f}_{ij}\left(\boldsymbol{z}_{ij}^{t}\right)\right\|_{2}^{2} \leq \frac{2}{\eta}\left(\tilde{f}_{ij}\left(\boldsymbol{z}_{ij}^{t}\right)-\mathbb{E}[\tilde{f}_{ij}\left(\boldsymbol{z}_{ij}^{t+1}\right)]\right)+\underbrace{\left\|\frac{\mathbb{E}[\boldsymbol{z}_{ij}^{t+1}]-\boldsymbol{z}_{ij}^{t}}{\eta}+\nabla \tilde{f}_{ij}\left(\boldsymbol{z}_{ij}^{t}\right)\right\|_{2}^{2}}_{=: \mathcal{T}}
$$

$$
+\frac{L \eta \sigma^{2}}{2}+\frac{2}{\eta} \frac{L}{2} \mathcal{T}_{ij}.
$$

(7)

The $\mathcal{T}$ can be upper bounded by expanding $z_{i}^{t+1}$ with (2)

$$
\begin{aligned}
\mathcal{T} :=& \left\|\frac{\mathbb{E}[\boldsymbol{z}_{ij}^{t+1}]-\boldsymbol{z}_{ij}^{t}}{\eta}+\nabla \tilde{f}_{ij}\left(\boldsymbol{z}_{ij}^{t}\right)\right\|_{2}^{2} \\
=& \left\|\nabla \tilde{f}_{ij}\left(\boldsymbol{z}_{ij}^{t}\right)-\frac{1}{2}\left(\nabla f_{i}(\boldsymbol{x}_{i}^{t})+\nabla f_{j}(\boldsymbol{x}_{j}^{t})\right)-\frac{\rho}{2}\left(\sum_{k=1}^{n} \mathbb{E}_{h}[w_{ik}^{t+1}](\boldsymbol{x}_{i}^{t}-\boldsymbol{x}_{k}^{t})+\sum_{k=1}^{n} \mathbb{E}_{h}[w_{jk}^{t+1}](\boldsymbol{x}_{j}^{t}-\boldsymbol{x}_{k}^{t})\right)\right\|_{2}^{2} \\
\leq& 3 \underbrace{\left\|\nabla \tilde{f}_{ij}\left(\boldsymbol{z}_{ij}^{t}\right)-\frac{1}{2}\left(\nabla f_{i}(\boldsymbol{x}_{i}^{t})+\nabla f_{j}(\boldsymbol{x}_{j}^{t})\right)\right\|_{2}^{2}}_{\mathcal{T}_{1}}+3 \underbrace{\left\|\frac{\rho}{2}\left(\sum_{k=1}^{n} w_{ik}^{\star}(\boldsymbol{x}_{i}^{t}-\boldsymbol{x}_{k}^{t})+\sum_{k=1}^{n} w_{jk}^{\star}(\boldsymbol{x}_{j}^{t}-\boldsymbol{x}_{k}^{t})\right)\right\|_{2}^{2}}_{\mathcal{T}_{2}} \\
& +3 \underbrace{\left\|\frac{\rho}{2}\left(\sum_{k=1}^{n}(\mathbb{E}_{h}[w_{ik}^{t+1}]-w_{ik}^{\star})(\boldsymbol{x}_{i}^{t}-\boldsymbol{x}_{k}^{t})+\sum_{k=1}^{n}(\mathbb{E}_{h}[w_{jk}^{t+1}]-w_{jk}^{\star})(\boldsymbol{x}_{j}^{t}-\boldsymbol{x}_{k}^{t})\right)\right\|_{2}^{2}}_{\mathcal{T}_{3}}.
\end{aligned}
$$

**Bound $\mathcal{T}_{1}$:** Use L-smoothness of $f_{i}$ and $f_{j}$. Take expectation with respect to $\boldsymbol{g}_{i}^{t}$ and $\boldsymbol{g}_{j}^{t}$ which are unbiased estimates of $\nabla f_{i}(\boldsymbol{x}_{i}^{t})$ and $\nabla f_{j}(\boldsymbol{x}_{j}^{t})$

$$
\begin{aligned}
\mathcal{T}_{1} =& \left\|\nabla \tilde{f}_{ij}\left(\boldsymbol{z}_{ij}^{t}\right)-\frac{1}{2}\left(\nabla f_{i}(\boldsymbol{x}_{i}^{t})+\nabla f_{j}(\boldsymbol{x}_{j}^{t})\right)\right\|_{2}^{2} \\
\leq& \frac{L^{2}}{2}\left\|\boldsymbol{z}_{ij}^{t}-\boldsymbol{x}_{i}^{t}\right\|_{2}^{2}+\frac{L^{2}}{2}\left\|\boldsymbol{z}_{ij}^{t}-\boldsymbol{x}_{j}^{t}\right\|_{2}^{2} \\
=& \frac{L^{2}}{4}\left\|\boldsymbol{x}_{i}^{t}-\boldsymbol{x}_{j}^{t}\right\|_{2}^{2}.
\end{aligned}
$$

**Bound $\mathcal{T}_{2}$:** Use Cauchy-Schwarz inequality and $\mathcal{C}$ has a cluster size of $c$

$$
\mathcal{T}_{2} \leq \frac{c \rho^{2}}{2}\left(\sum_{k=1}^{n} w_{ik}^{\star}\left\|\boldsymbol{x}_{i}^{t}-\boldsymbol{x}_{k}^{t}\right\|_{2}^{2}+\sum_{k=1}^{n} w_{jk}^{\star}\left\|\boldsymbol{x}_{j}^{t}-\boldsymbol{x}_{k}^{t}\right\|_{2}^{2}\right).
$$

**Bound $\mathcal{T}_{3}$:** Use Cauchy-Schwarz inequality

$$
\mathcal{T}_{3} \leq \frac{n \rho^{2}}{2}\left(\sum_{k=1}^{n}|\mathbb{E}_{h}[w_{ik}^{t+1}]-w_{ik}^{\star}|^{2}\left\|\boldsymbol{x}_{i}^{t}-\boldsymbol{x}_{k}^{t}\right\|_{2}^{2}+\sum_{k=1}^{n}|\mathbb{E}_{h}[w_{jk}^{t+1}]-w_{jk}^{\star}|^{2}\left\|\boldsymbol{x}_{j}^{t}-\boldsymbol{x}_{k}^{t}\right\|_{2}^{2}\right).
$$

Sum up $\left\|\nabla \tilde{f}_{ij}\left(\boldsymbol{z}_{ij}^{t}\right)\right\|_{2}^{2}$ for all of $i, j$ in the same cluster $\mathcal{C}$ for (7) yields

$$
\begin{aligned}
\sum_{i,j\in\mathcal{C}}\left\|\nabla \tilde{f}_{ij}\left(\boldsymbol{z}_{ij}^{t}\right)\right\|_{2}^{2} \leq & \frac{2}{\eta}\sum_{i,j\in\mathcal{C}}\left(\tilde{f}_{ij}\left(\boldsymbol{z}_{ij}^{t}\right)-\mathbb{E}[\tilde{f}_{ij}\left(\boldsymbol{z}_{ij}^{t+1}\right)]\right)+\frac{3L^{2}}{4}\sum_{i,j\in\mathcal{C}}\left\|\boldsymbol{x}_{i}^{t}-\boldsymbol{x}_{j}^{t}\right\|_{2}^{2} \\
& +3c^{2}\rho^{2}\sum_{i,j\in\mathcal{C}}\left\|\boldsymbol{x}_{i}^{t}-\boldsymbol{x}_{j}^{t}\right\|_{2}^{2}+3nc\rho^{2}\sum_{i\in\mathcal{C}}\sum_{k=1}^{n}|\mathbb{E}_{h}[w_{ik}^{t+1}]-w_{ik}^{\star}|^{2}\left\|\boldsymbol{x}_{i}^{t}-\boldsymbol{x}_{k}^{t}\right\|_{2}^{2} \\
& +\frac{c^{2}L\eta\sigma^{2}}{2}+\frac{2}{\eta}\frac{L}{2}\sum_{i,j\mathcal{C}}\mathcal{T}_{ij} \\
\leq & \frac{2}{\eta}\sum_{i,j\in\mathcal{C}}\left(\tilde{f}_{ij}\left(\boldsymbol{z}_{ij}^{t}\right)-\tilde{f}_{ij}\left(\boldsymbol{z}_{ij}^{t+1}\right)\right)+\left(\frac{3L^{2}}{4}+3c^{2}\rho^{2}\right)\sum_{i,j\in\mathcal{C}}\left\|\boldsymbol{x}_{i}^{t}-\boldsymbol{x}_{j}^{t}\right\|_{2}^{2} \\
& +3nc\rho^{2}\sum_{i\in\mathcal{C}}\sum_{k=1}^{n}|\mathbb{E}_{h}[w_{ik}^{t+1}]-w_{ik}^{\star}|^{2}\left\|\boldsymbol{x}_{i}^{t}-\boldsymbol{x}_{k}^{t}\right\|_{2}^{2}+\frac{c^{2}L\eta\sigma^{2}}{2}+\frac{2}{\eta}\frac{L}{2}\sum_{i,j\mathcal{C}}\mathcal{T}_{ij}.
\end{aligned}
$$
(8)

Now we expand $\mathcal{T}_{ij}$ as follows

$$
\begin{aligned}
\mathcal{T}_{ij} =& \mathbb{E}_{h}\mathbb{E}_{g}\left[\|z_{ij}^{t+1}-\mathbb{E}_{h}[z_{ij}^{t+1}]\|_{2}^{2}\right] \\
=& \mathbb{E}_{h}\mathbb{E}_{g}\left[\left\|\frac{\eta}{2}(g_{i}^{t}+g_{j}^{t})+\frac{\eta\rho}{2}\sum_{k=1}^{n}(w_{ik}^{t+1}(x_{i}^{t}-x_{k}^{t})+w_{jk}^{t+1}(x_{j}^{t}-x_{k}^{t}))\right.\right. \\
& \left.\left.-\mathbb{E}_{h}\left[\frac{\eta}{2}(g_{i}^{t}+g_{j}^{t})+\frac{\eta\rho}{2}\sum_{k=1}^{n}(w_{ik}^{t+1}(x_{i}^{t}-x_{k}^{t})+w_{jk}^{t+1}(x_{j}^{t}-x_{k}^{t}))\right]\right\|_{2}^{2}\right] \\
=& \mathbb{E}_{h}\left[\left\|\frac{\eta\rho}{2}\sum_{k=1}^{n}((w_{ik}^{t+1}-\mathbb{E}_{h}[w_{ik}^{t+1}])(x_{i}^{t}-x_{k}^{t})+(w_{jk}^{t+1}-\mathbb{E}_{h}[w_{jk}^{t+1}])(x_{j}^{t}-x_{k}^{t}))\right\|_{2}^{2}\right] \\
=& \frac{\eta^{2}\rho^{2}}{4}\sum_{k\neq i,j}\left(\mathbb{E}_{h}\left[\left\|w_{ik}^{t+1}-\mathbb{E}_{h}[w_{ik}^{t+1}]\right\|_{2}^{2}\right]\|x_{i}^{t}-x_{k}^{t}\|_{2}^{2}+\mathbb{E}_{h}\left[\left\|w_{jk}^{t+1}-\mathbb{E}_{h}[w_{jk}^{t+1}]\right\|_{2}^{2}\right]\|x_{j}^{t}-x_{k}^{t}\|_{2}^{2}\right)
\end{aligned}
$$

where we use the independence of random variables in the last equality.

Average the above equality over i, j$\in\mathcal{C}$ yields

$$
\begin{aligned}
\frac{1}{c^{2}}\sum_{ij}\mathcal{T}_{ij} =& \frac{\eta^{2}\rho^{2}(c-2)}{2c^{2}}\sum_{i,j}\mathbb{E}_{h}\left[\left\|w_{ij}^{t+1}-\mathbb{E}_{h}[w_{ij}^{t+1}]\right\|_{2}^{2}\right]\|x_{i}^{t}-x_{j}^{t}\|_{2}^{2} \\
\leq & \frac{\eta^{2}\rho^{2}(c-2)}{2c^{2}}\frac{4\sigma^{2}}{b}\sum_{i,j}\|x_{i}^{t}-x_{j}^{t}\|_{2}^{2}.
\end{aligned}
$$

Therefore, (8) is upper bounded with

$$\sum_{i,j\in\mathcal{C}}\left\|\nabla\tilde{f}_{ij}\left(\boldsymbol{z}_{ij}^t\right)\right\|_2^2 \leq \frac{2}{\eta}\sum_{i,j\in\mathcal{C}}\left(\tilde{f}_{ij}\left(\boldsymbol{z}_{ij}^t\right)-\mathbb{E}[\tilde{f}_{ij}\left(\boldsymbol{z}_{ij}^{t+1}\right)]\right)+\left(\frac{3L^2}{4}+3c^2\rho^2\right)\sum_{i,j\in\mathcal{C}}\left\|\boldsymbol{x}_i^t-\boldsymbol{x}_j^t\right\|_2^2$$

$$+3nc\rho^2\sum_{i\in\mathcal{C}}\sum_{k=1}^n|\mathbb{E}_h[w_{ik}^{t+1}]-w_{ik}^\star|^2\left\|\boldsymbol{x}_i^t-\boldsymbol{x}_k^t\right\|_2^2+\frac{c^2L\eta\sigma^2}{2}$$

$$+\frac{2}{\eta}\frac{L}{2}\frac{\eta^2\rho^2(c-2)}{2}\frac{4\sigma^2}{b}\sum_{i,j}\|x_i^t-x_j^t\|_2^2$$

$$=\frac{2}{\eta}\sum_{i,j\in\mathcal{C}}\left(\tilde{f}_{ij}\left(\boldsymbol{z}_{ij}^t\right)-\mathbb{E}[\tilde{f}_{ij}\left(\boldsymbol{z}_{ij}^{t+1}\right)]\right)$$

$$+\left(\frac{3L^2}{4}+3c^2\rho^2+\frac{L\eta\rho^2(c-2)2\sigma^2}{b}\right)\sum_{i,j\in\mathcal{C}}\left\|\boldsymbol{x}_i^t-\boldsymbol{x}_j^t\right\|_2^2$$

$$+3nc\rho^2\sum_{i\in\mathcal{C}}\sum_{k=1}^n|\mathbb{E}_h[w_{ik}^{t+1}]-w_{ik}^\star|^2\left\|\boldsymbol{x}_i^t-\boldsymbol{x}_k^t\right\|_2^2+\frac{c^2L\eta\sigma^2}{2}.$$

$\square$

**Lemma 4.** *Suppose* $M_{ij}\leq\frac{1}{5}$. *Let* $\rho\geq\frac{\sqrt{3}L}{c}$, $b\geq\frac{2}{c^2}2L\eta(c-2)\sigma^2$, *and* $\eta\leq\frac{1}{2\rho c}\leq\frac{1}{2\sqrt{3}L}$ *then*

$$\frac{1}{c^2}\sum_{i,j\in\mathcal{C}}\mathbb{E}\left[\left\|\boldsymbol{x}_i^{t+1}-\boldsymbol{x}_j^{t+1}\right\|_2^2\right]\leq(1-\eta\rho c)\frac{1}{c^2}\sum_{i,j\in\mathcal{C}}\left\|\boldsymbol{x}_i^t-\boldsymbol{x}_j^t\right\|_2^2$$

$$+\frac{5n\eta^2\rho^2}{L}\frac{1}{cn}\sum_{i\in\mathcal{C}}\sum_{k=1}^n\mathbb{E}_h\left[|w_{ik}^{t+1}-w_{ik}^\star|^2\right]\left\|\boldsymbol{x}_i^t-\boldsymbol{x}_k^t\right\|_2^2$$

$$+\frac{6M_{ij}^2}{\rho c}\frac{1}{c^2}\sum_{i,j\in\mathcal{C}}\left(\tilde{f}_{ij}\left(\boldsymbol{z}_{ij}^t\right)-\mathbb{E}\left[\tilde{f}_{ij}\left(\boldsymbol{z}_{ij}^{t+1}\right)\right]\right)$$

$$+\left(\frac{1}{c^2}+3\eta LM_{ij}^2\right)\frac{\eta\sigma^2}{2\rho c}.$$

*Proof.* Expand $\boldsymbol{x}_i^{t+1}-\boldsymbol{x}_j^{t+1}$ with (2)

$$\boldsymbol{x}_i^{t+1}-\boldsymbol{x}_j^{t+1}=\boldsymbol{x}_i^t-\boldsymbol{x}_j^t-\eta\left(\boldsymbol{g}_i^t-\boldsymbol{g}_j^t\right)-\eta\rho\sum_{k=1}^n\left(w_{ik}^{t+1}(\boldsymbol{x}_i^t-\boldsymbol{x}_k^t)-w_{jk}^{t+1}(\boldsymbol{x}_j^t-\boldsymbol{x}_k^t)\right).$$

As $i$ and $j$ belong to the same cluster (i.e., $w_{ij}^\star=1$), we add $\pm2\eta\rho\sum_{k=1}^n w_{ik}^\star(\boldsymbol{x}_i^t-\boldsymbol{x}_k^t)$

$$\boldsymbol{x}_i^{t+1}-\boldsymbol{x}_j^{t+1}=(1-2\eta\rho c)(\boldsymbol{x}_i^t-\boldsymbol{x}_j^t)-\eta\left(\boldsymbol{g}_i^t-\boldsymbol{g}_j^t\right)$$

$$-\eta\rho\sum_{k=1}^n\left((w_{ik}^{t+1}-w_{ik}^\star)(\boldsymbol{x}_i^t-\boldsymbol{x}_k^t)-(w_{jk}^{t+1}-w_{jk}^\star)(\boldsymbol{x}_j^t-\boldsymbol{x}_k^t)\right).$$

Compute the norm of $\boldsymbol{x}_i^{t+1}-\boldsymbol{x}_j^{t+1}$ and choose $\eta\rho\leq\frac{1}{2c}$ to use Jensen's inequality

$$\left\|\mathbb{E}_g[\boldsymbol{x}_i^{t+1}-\boldsymbol{x}_j^{t+1}]\right\|_2^2\leq(1-2\eta\rho c)\left\|\boldsymbol{x}_i^t-\boldsymbol{x}_j^t\right\|_2^2$$

$$+2\eta\rho c\left\|\frac{1}{2c}\sum_{k=1}^n\left((w_{ik}^{t+1}-w_{ik}^\star)(\boldsymbol{x}_i^t-\boldsymbol{x}_k^t)-(w_{jk}^{t+1}-w_{jk}^\star)(\boldsymbol{x}_j^t-\boldsymbol{x}_k^t)\right)\right.$$

$$\left.+\frac{1}{2\rho c}\left(\nabla f_i(\boldsymbol{x}_i^t)-\nabla f_j(\boldsymbol{x}_j^t)\right)\right\|_2^2.$$

Expand the right-hand side with Cauchy-Schwarz inequality

$$
\begin{aligned}
\left\|\mathbb{E}_g[\boldsymbol{x}_i^{t+1} - \boldsymbol{x}_j^{t+1}]\right\|_2^2 \leq &(1 - 2\eta\rho c)\|\boldsymbol{x}_i^t - \boldsymbol{x}_j^t\|_2^2 \\
&+ 4\eta\rho c\left\|\frac{1}{2c}\sum_{k=1}^n \left((w_{ik}^{t+1} - w_{ik}^\star)(\boldsymbol{x}_i^t - \boldsymbol{x}_k^t) - (w_{jk}^{t+1} - w_{jk}^\star)(\boldsymbol{x}_j^t - \boldsymbol{x}_k^t)\right)\right\|_2^2 \\
&+ 4\eta\rho c\left\|\frac{1}{2\rho c}\left(\nabla f_i(\boldsymbol{x}_i^t) - \nabla f_j(\boldsymbol{x}_j^t)\right)\right\|_2^2 \\
\leq &(1 - 2\eta\rho c)\|\boldsymbol{x}_i^t - \boldsymbol{x}_j^t\|_2^2 + 8\eta\rho c\left\|\frac{1}{2c}\sum_{k=1}^n (w_{ik}^{t+1} - w_{ik}^\star)(\boldsymbol{x}_i^t - \boldsymbol{x}_k^t)\right\|_2^2 \\
&+ 8\eta\rho c\left\|\frac{1}{2c}\sum_{k=1}^n (w_{jk}^{t+1} - w_{jk}^\star)(\boldsymbol{x}_j^t - \boldsymbol{x}_k^t)\right\|_2^2 \\
&+ 4\eta\rho c\left\|\frac{1}{2\rho c}\left(\nabla f_i(\boldsymbol{x}_i^t) - \nabla f_j(\boldsymbol{x}_j^t)\right)\right\|_2^2 \\
\leq &(1 - 2\eta\rho c)\|\boldsymbol{x}_i^t - \boldsymbol{x}_j^t\|_2^2 + \frac{2n\eta\rho}{c}\sum_{k=1}^n |w_{ik}^{t+1} - w_{ik}^\star|^2\|\boldsymbol{x}_i^t - \boldsymbol{x}_k^t\|_2^2 \\
&+ \frac{2n\eta\rho}{c}\sum_{k=1}^n |w_{jk}^{t+1} - w_{jk}^\star|^2\|\boldsymbol{x}_j^t - \boldsymbol{x}_k^t\|_2^2 + \frac{\eta}{\rho c}\underbrace{\|\nabla f_i(\boldsymbol{x}_i^t) - \nabla f_j(\boldsymbol{x}_j^t)\|_2^2}_{=:\mathcal{T}}.
\end{aligned}
$$

The last term $\mathcal{T}$ can be upper bounded by adding $\pm\nabla f_i\left(\boldsymbol{z}_{ij}^t\right) \pm \nabla f_j\left(\boldsymbol{z}_{ij}^t\right)$ and use L-smoothness assumption (A1) of $f_i$ and that $i$, $j$ belong to the same cluster (A4)

$$
\begin{aligned}
\mathcal{T} = &\left\|\nabla f_i(\boldsymbol{x}_i^t) \pm \nabla f_i\left(\boldsymbol{z}_{ij}^t\right) \pm \nabla f_j\left(\boldsymbol{z}_{ij}^t\right) - \nabla f_j(\boldsymbol{x}_j^t)\right\|_2^2 \\
\leq &3\left\|\nabla f_i(\boldsymbol{x}_i^t) - \nabla f_i\left(\boldsymbol{z}_{ij}^t\right)\right\|_2^2 + 3\left\|\nabla f_i\left(\boldsymbol{z}_{ij}^t\right) - \nabla f_j\left(\boldsymbol{z}_{ij}^t\right)\right\|_2^2 \\
&+ 3\left\|\nabla f_j(\boldsymbol{x}_j^t) - \nabla f_j\left(\boldsymbol{z}_{ij}^t\right)\right\|_2^2 \\
\leq &\frac{3L^2}{2}\|\boldsymbol{x}_i^t - \boldsymbol{x}_j^t\|_2^2 + 3M_{ij}^2\left\|\nabla f_i\left(\boldsymbol{z}_{ij}^t\right) + \nabla f_j\left(\boldsymbol{z}_{ij}^t\right)\right\|_2^2 \\
= &\frac{3L^2}{2}\|\boldsymbol{x}_i^t - \boldsymbol{x}_j^t\|_2^2 + 3M_{ij}^2\left\|\nabla \tilde{f}_{ij}(\boldsymbol{z}_{ij}^t)\right\|_2^2.
\end{aligned}
$$

Note that $\mathbb{E}[\|\boldsymbol{x}_i^{t+1} - \boldsymbol{x}_j^{t+1}\|_2^2] = \eta^2\,\mathbb{E}[\|\boldsymbol{g}_i^t - \boldsymbol{g}_j^t - \mathbb{E}[\boldsymbol{g}_i^t - \boldsymbol{g}_j^t]\|_2^2] + \mathbb{E}_h\|\mathbb{E}_g[\boldsymbol{x}_i^{t+1} - \boldsymbol{x}_j^{t+1}]\|_2^2$ and independence between randomness on worker $i$ and $j$ where expectation is take to all of the randomness until time $t$

By averaging for all $i, j \in \mathcal{C}$

$$
\begin{aligned}
\frac{1}{c^2}\sum_{i,j\in\mathcal{C}}\mathbb{E}\left[\|\boldsymbol{x}_i^{t+1} - \boldsymbol{x}_j^{t+1}\|_2^2\right] \leq &\left(1 - 2\eta\rho c + \frac{3\eta L^2}{2\rho c}\right)\frac{1}{c^2}\sum_{i,j\in\mathcal{C}}\|\boldsymbol{x}_i^t - \boldsymbol{x}_j^t\|_2^2 \\
&+ \frac{4n^2\eta\rho}{c}\frac{1}{cn}\sum_{i\in\mathcal{C}}\sum_{k=1}^n\mathbb{E}_h\left[|w_{ik}^{t+1} - w_{ik}^\star|^2\right]\|\boldsymbol{x}_i^t - \boldsymbol{x}_k^t\|_2^2 \\
&+ \frac{\eta}{\rho c}2\sigma^2 + \frac{3\eta M_{ij}^2}{\rho c}\frac{1}{c^2}\sum_{i,j\in\mathcal{C}}\left\|\nabla\tilde{f}_{ij}(\boldsymbol{z}_{ij}^t)\right\|_2^2.
\end{aligned}
$$

Use the previous Lemma 3 to bound $\sum_{i,j\in\mathcal{C}}\left\|\nabla\tilde{f}_{ij}(\boldsymbol{z}_{ij}^t)\right\|_2^2$

$$\frac{1}{c^2}\sum_{i,j\in\mathcal{C}}\mathbb{E}\left[\left\|\boldsymbol{x}_i^{t+1}-\boldsymbol{x}_j^{t+1}\right\|_2^2\right]$$

$$\leq\left(1-2\eta\rho c+\frac{3\eta L^2}{2\rho c}\right)\frac{1}{c^2}\sum_{i,j\in\mathcal{C}}\left\|\boldsymbol{x}_i^t-\boldsymbol{x}_j^t\right\|_2^2+\frac{4n^2\eta\rho}{c}\frac{1}{cn}\sum_{i\in\mathcal{C}}\sum_{k=1}^n\mathbb{E}_h\left[|w_{ik}^{t+1}-w_{ik}^\star|^2\right]\left\|\boldsymbol{x}_i^t-\boldsymbol{x}_k^t\right\|_2^2$$

$$+\frac{\eta\sigma^2}{2\rho c}\frac{1}{c^2}+\frac{3\eta M_{ij}^2}{\rho c}\left(\frac{2}{\eta}\frac{1}{c^2}\sum_{i,j\in\mathcal{C}}\left(\tilde{f}_{ij}\left(\boldsymbol{z}_{ij}^t\right)-\mathbb{E}\left[\tilde{f}_{ij}\left(\boldsymbol{z}_{ij}^{t+1}\right)\right]\right)+D_1\frac{1}{c^2}\sum_{i,j\in\mathcal{C}}\left\|\boldsymbol{x}_i^t-\boldsymbol{x}_j^t\right\|_2^2\right)$$

$$+\frac{3\eta M_{ij}^2}{\rho c}\left(3n^2\rho^2\frac{1}{cn}\sum_{i\in\mathcal{C}}\sum_{k=1}^n\mathbb{E}_h\left[|w_{ik}^{t+1}-w_{ik}^\star|^2\right]\left\|\boldsymbol{x}_i^t-\boldsymbol{x}_k^t\right\|_2^2+\frac{L\eta\sigma^2}{2}\right).$$

Rearrange the terms

$$\frac{1}{c^2}\sum_{i,j\in\mathcal{C}}\mathbb{E}\left[\left\|\boldsymbol{x}_i^{t+1}-\boldsymbol{x}_j^{t+1}\right\|_2^2\right]$$

$$\leq\left(1-2\eta\rho c+\frac{3\eta L^2}{2\rho c}+\frac{3\eta M_{ij}^2}{\rho c}D_1\right)\frac{1}{c^2}\sum_{i,j\in\mathcal{C}}\left\|\boldsymbol{x}_i^t-\boldsymbol{x}_j^t\right\|_2^2$$

$$+\left(\frac{4n^2\eta\rho}{c}+\frac{3\eta M_{ij}^2}{\rho c}3n^2\rho^2\right)\frac{1}{cn}\sum_{i\in\mathcal{C}}\sum_{k=1}^n\mathbb{E}_h\left[|w_{ik}^{t+1}-w_{ik}^\star|^2\right]\left\|\boldsymbol{x}_i^t-\boldsymbol{x}_k^t\right\|_2^2$$

$$+\frac{3\eta M_{ij}^2}{\rho c}\frac{2}{\eta}\frac{1}{c^2}\sum_{i,j\in\mathcal{C}}\left(\tilde{f}_{ij}\left(\boldsymbol{z}_{ij}^t\right)-\tilde{f}_{ij}\left(\boldsymbol{z}_{ij}^{t+1}\right)\right)+\frac{\eta\sigma^2}{2\rho c}\frac{1}{c^2}+\frac{3\eta M_{ij}^2}{\rho c}\frac{L\eta\sigma^2}{2}.$$

We would like to achieve $\left(1-2\eta\rho c+\frac{3\eta L^2}{2\rho c}+\frac{3\eta M_{ij}^2}{\rho c}D_1\right)\leq(1-\eta\rho c)$ by

- letting $\rho\geq\frac{\sqrt{3}L}{c}$, s.t. $\frac{3\eta L^2}{2\rho c}\leq\frac{\eta\rho c}{2}$.

- letting $b\geq\frac{2}{c^2}2L\eta(c-2)\sigma^2$ and $\rho\geq\frac{\sqrt{3}L}{c}$ and $M_{ij}\leq\frac{1}{5}$, the following inequality hold true

$$\frac{3\eta M_{ij}^2}{\rho c}\left(\frac{3L^2}{4}+3c^2\rho^2+\frac{L\eta\rho^2(c-2)2\sigma^2}{b}\right)\leq\frac{3\eta M_{ij}^2}{\rho c}\frac{15}{4}\rho^2c^2\leq\frac{45}{4}\rho c\eta M_{ij}^2\leq\frac{1}{2}\eta\rho c.$$

Using the same requirement that $\rho\geq\frac{\sqrt{3}L}{c}$ and $M_{ij}\leq\frac{1}{5}$

$$\frac{4n^2\eta\rho}{c}+\frac{3\eta M_{ij}^2}{\rho c}3n^2\rho^2\leq\frac{4n^2\eta\rho^2}{\sqrt{3}L}+\frac{3\sqrt{3}\eta M_{ij}^2n^2\rho^2}{L}\leq\frac{5\eta n^2\rho^2}{L}.$$

The upper bound of $1/c^2\sum_{i,j\in\mathcal{C}}\left\|\boldsymbol{x}_i^{t+1}-\boldsymbol{x}_j^{t+1}\right\|_2^2$ can be simplified to

$$\frac{1}{c^2}\sum_{i,j\in\mathcal{C}}\mathbb{E}\left[\left\|\boldsymbol{x}_i^{t+1}-\boldsymbol{x}_j^{t+1}\right\|_2^2\right]\leq(1-\eta\rho c)\frac{1}{c^2}\sum_{i,j\in\mathcal{C}}\left\|\boldsymbol{x}_i^t-\boldsymbol{x}_j^t\right\|_2^2$$

$$+\frac{5n\eta^2\rho^2}{L}\frac{1}{cn}\sum_{i\in\mathcal{C}}\sum_{k=1}^n\mathbb{E}_h\left[|w_{ik}^{t+1}-w_{ik}^\star|^2\right]\left\|\boldsymbol{x}_i^t-\boldsymbol{x}_k^t\right\|_2^2$$

$$+\frac{6M_{ij}^2}{\rho c}\frac{1}{c^2}\sum_{i,j\in\mathcal{C}}\left(\tilde{f}_{ij}\left(\boldsymbol{z}_{ij}^t\right)-\mathbb{E}\left[\tilde{f}_{ij}\left(\boldsymbol{z}_{ij}^{t+1}\right)\right]\right)$$

$$+\left(\frac{1}{c^2}+3\eta LM_{ij}^2\right)\frac{\eta\sigma^2}{2\rho c}.$$

$\square$

## A.2 Proof of Theorem I

*Proof.* Given Lemma 4 and average over time $t = 0$ over $T-1$ and take expectation to all randomness throughout training

$$\frac{1}{c^2 T} \sum_{t=0}^{T-1} \sum_{i,j \in \mathcal{C}} \mathbb{E}\left[\left\|\boldsymbol{x}_i^{t+1} - \boldsymbol{x}_j^{t+1}\right\|_2^2\right] \leq (1 - \eta\rho c) \frac{1}{c^2 T} \sum_{t=0}^{T-1} \sum_{i,j \in \mathcal{C}} \mathbb{E}\left[\left\|\boldsymbol{x}_i^t - \boldsymbol{x}_j^t\right\|_2^2\right]$$

$$+ \frac{5n\eta^2\rho^2}{L} \frac{1}{Tcn} \sum_{t=0}^{T-1} \sum_{i \in \mathcal{C}} \sum_{k=1}^{n} \mathbb{E}\left[|w_{ik}^{t+1} - w_{ik}^\star|^2 \|\boldsymbol{x}_i^t - \boldsymbol{x}_k^t\|_2^2\right]$$

$$+ \frac{6M_{ij}^2}{\rho c} \frac{1}{Tc^2} \sum_{t=0}^{T-1} \sum_{i,j \in \mathcal{C}} \left(\mathbb{E}\left[\tilde{f}_{ij}\left(\boldsymbol{z}_{ij}^t\right)\right] - \mathbb{E}\left[\tilde{f}_{ij}\left(\boldsymbol{z}_{ij}^{t+1}\right)\right]\right)$$

$$+ \left(\frac{1}{c^2} + 3\eta L M_{ij}^2\right) \frac{\eta\sigma^2}{2\rho c}.$$

Rearrange $\frac{1}{c^2 T} \sum_{t=0}^{T-1} \sum_{i,j \in \mathcal{C}} \mathbb{E}\left[\left\|\boldsymbol{x}_i^{t+1} - \boldsymbol{x}_j^{t+1}\right\|_2^2\right]$ yields

$$\frac{1}{c^2 T} \sum_{t=0}^{T-1} \sum_{i,j \in \mathcal{C}} \mathbb{E}\left[\left\|\boldsymbol{x}_i^{t+1} - \boldsymbol{x}_j^{t+1}\right\|_2^2\right] \leq \frac{5n\eta\rho}{Lc} \frac{1}{Tcn} \sum_{t=0}^{T-1} \sum_{i \in \mathcal{C}} \sum_{k=1}^{n} \mathbb{E}\left[|w_{ik}^t - w_{ik}^\star|^2 \|\boldsymbol{x}_i^t - \boldsymbol{x}_k^t\|_2^2\right]$$

$$+ \frac{6M_{ij}^2}{\eta\rho^2 c^2} \frac{1}{c^2 T} \sum_{t=0}^{T-1} \sum_{i,j \in \mathcal{C}} \mathbb{E}\left[\left(\tilde{f}_{ij}\left(\boldsymbol{z}_{ij}^t\right) - \tilde{f}_{ij}\left(\boldsymbol{z}_{ij}^{t+1}\right)\right)\right]$$

$$+ \left(\frac{1}{c^2} + 3\eta L M_{ij}^2\right) \frac{\sigma^2}{2\rho^2 c^2}.$$

Consider bounding $|w_{ik}^t - w_{ik}^\star|^2$ in two cases

**Case 1:** $w_{ik}^\star = 1$. Suppose $M_{ik} \in (0, 1)$, then $\|\nabla f_i(z_{ik}^t) - \nabla f_k(z_{ik}^t)\|_2^2 \leq M_{ij}^2 \|\nabla f_i(z_{ik}^t) + \nabla f_j(z_{ik}^t)\|_2^2$ implies

$$\langle \nabla f_i(z_{ik}^t), \nabla f_k(z_{ik}^t) \rangle \geq \frac{1 - M_{ik}^2}{2(1 + M_{ik}^2)} \left(\|\nabla f_i(z_{ik}^t)\|_2^2 + \|\nabla f_k(z_{ik}^t)\|_2^2\right) \geq 0.$$

then $w_{ik}^{t+1} = w_{ik}^\star = 1$ and therefore $|w_{ik}^{t+1} - w_{ik}^\star|^2 = 0$.

**Case 2:** $w_{ik}^\star = 0$. Suppose $\zeta_{ik}^2 \geq \|\nabla f_i(\boldsymbol{x}) + \nabla f_k(\boldsymbol{x})\|_2^2$ for all $\boldsymbol{x}$ then

$$\|\nabla f_i(\boldsymbol{z}_{ik}^t) + \nabla f_k(\boldsymbol{z}_{ik}^t)\|_2^2 = \|\nabla f_i(\boldsymbol{z}_{ik}^t)\|_2^2 + \|\nabla f_k(\boldsymbol{z}_{ik}^t)\|_2^2 + 2\langle \nabla f_i(\boldsymbol{z}_{ik}^t), \nabla f_k(\boldsymbol{z}_{ik}^t) \rangle$$
$$\geq \zeta_{ik}^2 + 2\langle \nabla f_i(\boldsymbol{z}_{ik}^t), \nabla f_k(\boldsymbol{z}_{ik}^t) \rangle$$

which means the inner product $\langle \nabla f_i(z_{ij}^t), \nabla f_j(z_{ij}^t) \rangle \leq 0$ is negative, i.e., $w_{ij}^{t+1} = 0 = w_{ij}^\star$. The above cases hold true for well initialized weights $\boldsymbol{x}^0$.

Then with lower bound assumption of $f_i$ and $f_j$ (A3)

$$\frac{1}{c^2 T} \sum_{t=0}^{T-1} \sum_{i,j \in \mathcal{C}} \mathbb{E}\left[\left\|\boldsymbol{x}_i^{t+1} - \boldsymbol{x}_j^{t+1}\right\|_2^2\right] \leq \frac{6M_{ij}^2}{\eta\rho^2 c^2} \frac{1}{c^2 T} \sum_{t=0}^{T-1} \sum_{i,j \in \mathcal{C}} \mathbb{E}\left[\left(\tilde{f}_{ij}\left(\boldsymbol{z}_{ij}^t\right) - \tilde{f}_{ij}\left(\boldsymbol{z}_{ij}^{t+1}\right)\right)\right]$$

$$+ \left(\frac{1}{c^2} + 3\eta L M_{ij}^2\right) \frac{\sigma^2}{2\rho^2 c^2}$$

$$\leq \frac{6M_{ij}^2}{\eta\rho^2 c^2} \frac{1}{c^2 T} \sum_{i,j \in \mathcal{C}} \mathbb{E}\left[\left(\tilde{f}_{ij}\left(\boldsymbol{z}_{ij}^0\right) - \tilde{f}_{ij}\left(\boldsymbol{z}_{ij}^T\right)\right)\right]$$

$$+ \left(\frac{1}{c^2} + 3\eta L M_{ij}^2\right) \frac{\sigma^2}{2\rho^2 c^2}$$

Minimize the upper bound through choosing $\eta$

$$\eta \leq \frac{2}{\sigma\sqrt{LT}}\sqrt{\frac{1}{c^2}\sum_{i,j\in\mathcal{C}}\left(\tilde{f}_{ij}\left(z_{ij}^0\right)-\tilde{f}_{ij}^\star\right)}$$

such that

$$\frac{1}{T}\sum_{t=0}^{T-1}\sum_{i,j\in\mathcal{C}}\mathbb{E}\left[\left\|x_i^{t+1}-x_j^{t+1}\right\|_2^2\right]\leq\frac{6M_{ij}^2}{\rho^2}\sqrt{\frac{L\sigma^2}{c^2T}\sum_{i,j\in\mathcal{C}}\left(\tilde{f}_{ij}\left(z_{ij}^0\right)-\tilde{f}_{ij}^\star\right)}. \tag{9}$$

By the result of Lemma 3

$$\frac{1}{T}\sum_{t=0}^{T-1}\sum_{i,j\in\mathcal{C}}\mathbb{E}\left[\left\|\nabla\tilde{f}_{ij}\left(z_{ij}^t\right)\right\|_2^2\right]\leq\frac{2}{\eta}\frac{1}{T}\sum_{i,j\in\mathcal{C}}\left(\tilde{f}_{ij}\left(z_{ij}^0\right)-\tilde{f}_{ij}^\star\right)+4c^2\rho^2\frac{1}{T}\sum_{t=0}^{T-1}\sum_{i,j\in\mathcal{C}}\mathbb{E}\left[\left\|x_i^t-x_j^t\right\|_2^2\right]+\frac{c^2L\eta\sigma^2}{2}$$

$$\leq 2c^2\sqrt{\frac{L\sigma^2}{c^2T}\sum_{i,j\in\mathcal{C}}\left(\tilde{f}_{ij}\left(z_{ij}^0\right)-\tilde{f}_{ij}^\star\right)}+4c^2\rho^2\frac{1}{T}\sum_{t=0}^{T-1}\sum_{i,j\in\mathcal{C}}\mathbb{E}\left[\left\|x_i^t-x_j^t\right\|_2^2\right]$$

$$\leq\left(2c^2+24c^2M_{ij}^2\right)\sqrt{\frac{L\sigma^2}{c^2T}\sum_{i,j\in\mathcal{C}}\left(\tilde{f}_{ij}\left(z_{ij}^0\right)-\tilde{f}_{ij}^\star\right)}$$

$$\leq 3c^2\sqrt{\frac{L\sigma^2}{c^2T}\sum_{i,j\in\mathcal{C}}\left(\tilde{f}_{ij}\left(z_{ij}^0\right)-\tilde{f}_{ij}^\star\right)}.$$

$\square$

### A.3  Proof of Corollary II

*Proof.* By adding $\|\nabla f_i(x)+\nabla f_j(x)\|_2^2$ on both sides of (5), and replace $x$ with $z_{ij}$ we have

$$2\left(\|\nabla f_i(z_{ij})\|_2^2+\|\nabla f_j(z_{ij})\|_2^2\right)\leq 4(1+M_{ij}^2)\left\|\nabla\tilde{f}_{ij}(z_{ij})\right\|_2^2\qquad\forall\,x\in\mathbb{R}^d. \tag{10}$$

Then using the upper bound of $M_{ij}<1/5$ from Theorem I, and average over $t$ and $i,j$ yields

$$\frac{1}{c^2T}\sum_{t=0}^{T-1}\sum_{i,j\in\mathcal{C}}\mathbb{E}\left[\|\nabla f_i\left(z_{ij}^t\right)\|_2^2\right]\leq\left(1+\frac{1}{25}\right)3\sqrt{\frac{L\sigma^2}{c^2T}\sum_{i,j\in\mathcal{C}}\left(\tilde{f}_{ij}\left(z_{ij}^0\right)-\tilde{f}_{ij}^\star\right)}. \tag{11}$$

By Cauchy-Schwarz inequality and $L$-Lipschitz smoothness, we have that

$$\frac{1}{c^2T}\sum_{t=0}^{T-1}\sum_{i,j\in\mathcal{C}}\|\nabla f_i\left(x_i^t\right)\|_2^2\leq\frac{1}{c^2T}\sum_{t=0}^{T-1}\sum_{i,j\in\mathcal{C}}\|\nabla f_i\left(z_{ij}^t\right)\|_2^2+\frac{1}{c^2T}\sum_{t=0}^{T-1}\sum_{i,j\in\mathcal{C}}\|\nabla f_i\left(z_{ij}^t\right)-\nabla f_i\left(x_i^t\right)\|_2^2$$

$$\leq\frac{1}{c^2T}\sum_{t=0}^{T-1}\sum_{i,j\in\mathcal{C}}\|\nabla f_i\left(z_{ij}^t\right)\|_2^2+\frac{L^2}{4}\frac{1}{c^2T}\sum_{t=0}^{T-1}\sum_{i,j\in\mathcal{C}}\|x_i^t-x_j^t\|_2^2.$$

Applying (9) and (11) to the upper bound of the above inequality

$$\frac{1}{c^2T}\sum_{t=0}^{T-1}\sum_{i,j\in\mathcal{C}}\|\nabla f_i\left(x_i^t\right)\|_2^2\leq\left(1+\frac{1}{25}\right)3\sqrt{\frac{L\sigma^2}{c^2T}\sum_{i,j\in\mathcal{C}}\left(\tilde{f}_{ij}\left(z_{ij}^0\right)-\tilde{f}_{ij}^\star\right)}$$

$$+\frac{L^2}{4}\frac{6M_{ij}^2}{\rho^2c^2}\sqrt{\frac{L\sigma^2}{c^2T}\sum_{i,j\in\mathcal{C}}\left(\tilde{f}_{ij}\left(z_{ij}^0\right)-\tilde{f}_{ij}^\star\right)}.$$

As $\rho c\geq\sqrt{3}L$ and $M_{ij}<\frac{1}{5}$ as stated in Theorem I,

$$\frac{1}{c^2T}\sum_{t=0}^{T-1}\sum_{i,j\in\mathcal{C}}\|\nabla f_i\left(x_i^t\right)\|_2^2\leq 4\sqrt{\frac{L\sigma^2}{c^2T}\sum_{i,j\in\mathcal{C}}\left(\tilde{f}_{ij}\left(z_{ij}^0\right)-\tilde{f}_{ij}^\star\right)}.$$

$\square$

# B   Experimental Details

In Section 4, we present our results on two tasks with different properties. Here, we provide the full details of our experimental setup, alongside with additional experiments.

We first describe the setup for cross-device and cross-silo experiments: we use the fix batch size of 128 for cross-device, and cross-silo experiments on CIFAR-100. We tune each method for the optimal learning rate individually: we use learning rate of 0.1 for ditto, 0.05 for Federated Clustering (FC), and 0.01 for all other methods. For Ditto, we use the hyper-parameter of $\lambda = 1$ as recommended in their paper. For Federated Clustering, we use the ground truth number of clusters and size of clusters as the hyper-parameter. We also use the ground truth number of clusters for IFCA, and sample all the clients in cross-silo experiment. We reduce the sampling rate to 10% for the cross-device experiment to ensure scalability and fairness for comparison to other methods. For cross-silo experiments we employed a single NVIDIA V-100 GPU with 32GB memory, and moved to four NVIDIA V-100 GPUs with 32 GB memory for cross-device experiment. With this setup, running COBO for cross-silo and cross-device experiment takes 9 hours and 28 hours respectively.

For Language modeling experiment, we conducted the experiments with the learning rate of 0.002, batch size of 50, and 4 accumulation steps. Note that each agent only get a subset of the regarding language from Wiki-40B dataset, consisting of total of 800000 tokens. We also used the context length of 512, dropout rate of 0.1, and LoRA module with rank 4. Training is performed on a single NVIDIA A-100 GPU with 40GB memory. It takes 2.5 hours to run COBO for 500 iterations in this framework. We also use $\lambda = 0.1$ for Ditto which has higher performance for this experiment. There are 3 strategies proposed in [38] for collaborative fine-tuning, highlighting one as the most effective. However, this approach requires a portion of dataset to be shared among agents, which is incompatible with our experimental setup. Therefore, we use the second-best method in our experiments, which relies on validation performance.



Figure 5: Collaboration matrices learned by COBO at different stages of training for cross-device experiment with 80 clients. The diagonals are masked out. The oracle matrix is a block diagonal matrix, consisting of 10 blocks: two blocks of size 10, two blocks of size 9, and so on. The collaboration matrix of COBO already starts to look similar to oracle matrix within as low as 300 iterations. (1.5% of the total iterations)

For the cross-device experiment with 80 agents in Section 4.3, we present the accuracy curve in Figure 6. Our method outperform all other methods except the Oracle with a large margin. We can also observe the collaboration matrix of COBO in Figure 5. The clusters are learned with COBO efficiently.


Figure 6: Averaged accuracies of personalized models for cross-device federated learning with 80 clients. The "Oracle" denotes applying FedAvg to the clients having the data from the same classes of CIFAR-100 dataset.

Question: Does the paper discuss the limitations of the work performed by the authors?

Answer: [Yes]

Justification: We list limitations at the end of the main text.

3. **Theory Assumptions and Proofs**

Question: For each theoretical result, does the paper provide the full set of assumptions and a complete (and correct) proof?

Answer: [Yes]

Justification: The assumptions are given in Section 3.

4. **Experimental Result Reproducibility**

Question: Does the paper fully disclose all the information needed to reproduce the main experimental results of the paper to the extent that it affects the main claims and/or conclusions of the paper (regardless of whether the code and data are provided or not)?

Answer: [Yes]

Justification: The experiments are clearly defined, and all the necessary hyperparameters for reproducing them are presented in the Appendix.

5. **Open access to data and code**

Question: Does the paper provide open access to the data and code, with sufficient instructions to faithfully reproduce the main experimental results, as described in supplemental material?

Answer: [Yes]

Justification: The code is available in supplementary materials as well as on an anonymous Github repository, and we only used publicly available datasets in our experiments.

6. **Experimental Setting/Details**

Question: Does the paper specify all the training and test details (e.g., data splits, hyperparameters, how they were chosen, type of optimizer, etc.) necessary to understand the results?

Answer: [Yes]

Justification: The full training setup is described in the main text. The hyper-parameters are also available both for CoBo and other baselines either in the main text or appendix.

7. **Experiment Statistical Significance**

   Question: Does the paper report error bars suitably and correctly defined or other appropriate information about the statistical significance of the experiments?

   Answer: [Yes]

   Justification: We report proper confidence intervals for all experiments in our work.

8. **Experiments Compute Resources**

   Question: For each experiment, does the paper provide sufficient information on the computer resources (type of compute workers, memory, time of execution) needed to reproduce the experiments?

   Answer: [Yes]

   Justification: We explained the training framework thoroughly in appendix. All the mentioned information are discussed and we believe it is sufficient for reproducing the experiments.

9. **Code Of Ethics**

   Question: Does the research conducted in the paper conform, in every respect, with the NeurIPS Code of Ethics https://neurips.cc/public/EthicsGuidelines?

   Answer: [Yes]

   Justification: We preserve the NeurIPS Code of Ethics throughout the research project.

10. **Broader Impacts**

    Question: Does the paper discuss both potential positive societal impacts and negative societal impacts of the work performed?

    Answer: [NA]

    Justification: Not applicable.

11. **Safeguards**

    Question: Does the paper describe safeguards that have been put in place for responsible release of data or models that have a high risk for misuse (e.g., pretrained language models, image generators, or scraped datasets)?

    Answer: [NA]

    Justification: Note applicable.

12. **Licenses for existing assets**

    Question: Are the creators or original owners of assets (e.g., code, data, models), used in the paper, properly credited and are the license and terms of use explicitly mentioned and properly respected?

    Answer: [Yes]

    Justification: We have cited the datasets.

13. **New Assets**

    Question: Are new assets introduced in the paper well documented and is the documentation provided alongside the assets?

    Answer: [NA]

    Justification: Not applicable

14. **Crowdsourcing and Research with Human Subjects**

    Question: For crowdsourcing experiments and research with human subjects, does the paper include the full text of instructions given to participants and screenshots, if applicable, as well as details about compensation (if any)?

    Answer: [NA]

    Justification: This paper does not involve crowdsourcing experiments.

15. **Institutional Review Board (IRB) Approvals or Equivalent for Research with Human Subjects**

Question: Does the paper describe potential risks incurred by study participants, whether such risks were disclosed to the subjects, and whether Institutional Review Board (IRB) approvals (or an equivalent approval/review based on the requirements of your country or institution) were obtained?

Answer: [NA]

Justification: Not applicable.

