# OpenReview forum: "CoBo: Collaborative Learning via Bilevel Optimization"
_NeurIPS.cc/2024/Conference — NeurIPS 2024 poster_

### Official Review · Reviewer_4QtE · 2024-07-12

**Soundness:** 3
**Presentation:** 3
**Contribution:** 2
**Rating:** 5
**Confidence:** 3

**Summary:**

In this paper, the authors introduce a novel approach to collaborative learning by framing it as a bilevel optimization problem, enhancing the training efficacy of multiple clients. In conventional collaborative learning paradigms, clients engage in mutual model training through information exchange; however, pinpointing beneficial collaborators presents a challenge and can incur significant computational overhead. To address this, the authors model client selection and model training as two interrelated optimization issues, proposing the COBO algorithm—a scalable and elastic Stochastic Gradient Descent (SGD) type alternating optimization algorithm. Theoretically guaranteed for convergence, COBO demonstrates superior performance over popular personalization algorithms.

**Strengths:**

1.The authors formulate collaborative learning with an innovative bilevel optimization framework that capitalizes on the intrinsic collaborative structure, yielding more universally applicable solutions.

2.The authors introduce COBO, a scalable and elastic SGD-type alternating optimization algorithm that efficiently addresses the bilevel problem challenge, scaling with an increasing number of clients while maintaining elasticity in client count.

3.The authors establish theoretical convergence guarantees for COBO in collaborative learning scenarios with clustered client structures.

4.The authors demonstrate that COBO surpasses established personalized federated learning benchmarks, especially in heterogeneous federated learning environments and when fine-tuning Large Language Models (LLMs).

**Weaknesses:**

According to equation (1), the collaboration weights are updated by applying projected gradient descent with step size . However, an estimate of the step size  is not provided.

**Questions:**

1.Could you provide an estimate of the step size in equation (1)? (see weaknesses)

2.The paper mentions the application of the COBO algorithm in federated learning environments where privacy is a critical consideration. What privacy protection measures have been incorporated into the COBO algorithm to ensure the security of client data, and how is the balance between privacy protection and model performance achieved in the algorithm design?

3.The COBO algorithm and the DITTO algorithm are formulated within the framework of  bilevel optimization. Compared to the DITTO algorithm, what additional theoretical guarantees or advantages does COBO provide? Furthermore, does COBO demonstrate greater flexibility and robustness when dealing with diverse tasks?

**Limitations:**

1.The proof of the COBO algorithm proposed in the paper is based on some simplified theoretical assumptions, such as the use of full gradient information in the inner problem and the assumption that the minimization problem is solved exactly. These may not fully reflect the complexities encountered in practical applications.

2.In the cross-device federated learning experiment involving 80 clients, each algorithm was executed only once. This implies that the results may not fully reflect the robustness of the algorithms under different random seeds or data partitions.

---

> ### Author Rebuttal · Authors · 2024-08-07
>
> We thank the reviewer for their insightful questions and comments.
>
> **Regarding Question 1**
>
> We appreciate the reviewer's interest in the step size selection for the projected gradient descent in CoBo. We choose $O(1/LT)$ as the step size, one intuitive explanation is motivated by the similarity between our formulation and the Frank-Wolfe algorithm. As noted in the Frank-Wolfe literature, step sizes of the order $O(1/(t+2))$ are commonly used.
>
> **Regarding Question 2**
>
> We acknowledge the importance of privacy in federated learning and appreciate the reviewer's concerns. While we do not incorporate specific privacy measures into CoBo in this paper, we note that differential privacy can be combined with CoBo directly. However, achieving output privacy may require additional cryptographic tools. We consider this a promising direction for future work. We hope the reviewer will consider this a natural next step rather than a drawback of our paper.
>
> **Regarding Question 3**
>
> We are glad to highlight the key differences between Ditto and CoBo, as described in Section 2 (between lines 57 and 62). In scenarios where clients are drawn from diverse distributions, possibly with conflicting labels, CoBo's flexible formulation allows clients to identify fine-grained cluster structures, leading to better local models and improved performance. In contrast, Ditto's formulation may penalize the distance between local models and a meaningless global model, potentially harming local model performance. We believe this example illustrates the advantages of CoBo in handling diverse and complex client distributions.

---

### Official Review · Reviewer_H7Ja · 2024-07-13

**Soundness:** 2
**Presentation:** 3
**Contribution:** 3
**Rating:** 6
**Confidence:** 2

**Summary:**

In this paper, the authors model collaborative learning as a bilevel optimization problem, and propose CoBo, an SGD-type alternating optimization algorithm, to solve this problem. Theoretical convergence guarantees are provided, and experiments are conducted to evaluate the performance of CoBo.

**Strengths:**

In this paper, the authors model client-selection and model-training as two interconnected optimization problems, proposing a bilevel optimization problem for collaborative learning. They introduce CoBo, an SGD-type alternating optimization algorithm designed to address this problem. CoBo is proven to have theoretical convergence guarantees for collaborative learning with cluster structures. In the experiments, CoBo outperforms popular personalized federated learning baselines.

**Weaknesses:**

1. Although this paper provides theoretical convergence guarantees, the proof of Theorem I is given for a simplified scenario where the inner problem uses the full gradient, which is difficult to compute in practice.
2. This paper does not provide comparisons of the theoretical convergence performance between CoBo and state-of-the-art algorithms.
2. Repeat experiments are not conducted to ensure the robustness of the results.


Minor:
Line 120: "They leads to eventual" is not a full sentence.

**Questions:**

1. In Line 7 of Algorithm 1, why use this output rather than return $\\{ x_0^T,\cdots \cdots ,x_N^T \\}$ and $W^T$?
2. The collaborative learning scenario used in this paper doesn't involve a central server. Why do the authors only compare CoBo with personalized federated learning baselines? Why don't they also compare it with some fully decentralized algorithms?
3. The probability 1/n in Line 11 of Algorithm 1 is an important parameter. How does it influence the performance of CoBo?

**Limitations:**

The authors mention that the limitations (see Weaknesses 1 and 3) are due to time limitations. This suggests that the paper may not be fully ready for submission.

---

> ### Author Rebuttal · Authors · 2024-08-07
>
> We would like to thank you for your thoughtful reviews and suggestions on our paper. We appreciate your feedback and are happy to address the concerns raised.
>
> Regarding Weaknesses:
>
> **Theorem I proof:** We agree with the reviewer that using full gradients for the inner problem simplifies the proof. In practice, we still use mini-batch SGD as the unbiased estimate of the full gradient. We plan to update the proof in the future to reflect this practical consideration.
>
> **Theoretical convergence performance comparison:** We provide a comparison of the theoretical convergence performance between CoBo and state-of-the-art algorithms in our response. Specifically, we show that CoBo enjoys linear scalability with respect to cluster size, whereas federated clustering requires O(n^2) gradient computations per iteration. Ditto gives a coarse grained O(1/T)  convergence rate for local models.
>
> **Robustness of results:** We acknowledge the importance of repeat experiments for ensuring the robustness of our results. We have repeated the experiments, and the results are available in the global PDF file.
>
> **Regarding Minor Issues:**
> Line 120: We apologize for the mistake and will ensure that the sentence is complete in the final version of the paper.
>
> Questions:
>
>
> **Output of Algorithm 1:** Non-convex optimization usually tries to prove that the time-averaged gradient norm, such as $\frac{1}{T} \sum_{t=0}^{T-1} \lVert \nabla f(x^t) \rVert_2^2$, is upper bounded by a very small value. However, the theoretical results do not guarantee that the last iterate $x^T$ is also  upper bounded. Instead, the $\frac{1}{T} \sum_{t=0}^{T-1} \lVert \nabla f(x^t) \rVert_2^2$ can be seen as the expected gradient norm of a uniform randomly drawn $x^t$, therefore this drawn iterate has the upper bound in expectation. This is a well-established technique in the literature (see [Fang et al. 2018] for an example).
> Comparison with fully decentralized algorithms: We appreciate your suggestion to compare CoBo with fully decentralized algorithms. However, our baseline federated clustering algorithm is already fully decentralized and does not rely on a central server. In contrast, personalized federated learning with a central server can be seen as a fully-connected decentralized graph, which is a more challenging baseline to beat.
>
> **Influence of sampling rate:** You are correct that the sampling rate of 1/n is an important parameter. We have carefully chosen this value to minimize the pairwise computation overhead while preserving the quality of the solution.
>
> a. Complexity: Minimizing outer n objectives for $T$ iterations require computing n * T gradients. Naively computing pairwise gradients for the inner objective requires n^2 T gradients computation which dominate the complexity. With a sampling rate of O(1/n), the overall complexity of the bilevel formulation remain the same as single level optimization.
>
> b. Preserved quality: uniform randomly sampling O(1/n) of the edges do not harm the performance of CoBo. As the underlying connectivity matrix is fixed throughout training,each edge will be sampled $O(T/n)$ times to determine the connectivity. As the number of iterations $T$ is usually much larger than the number of clients, it is enough to determine the connectivity.
>
> Fang C, Li C J, Lin Z, et al. Spider: Near-optimal non-convex optimization via stochastic path-integrated differential estimator[J]. Advances in neural information processing systems, 2018, 31.

---

> > ### Comment · Reviewer_H7Ja · 2024-08-12
> >
> > Thank the authors for the rebuttal. Since Weakness 1 has not yet been addressed, I would like to keep my current score.

---

> > > ### Author Response · Authors · 2024-08-12
> > > **Addressing weakness 1**
> > >
> > > Our changes mainly applies to the proof of Lemma 3, starting from line 390 in the original submission. The current proof only consider the randomness of stochastic gradient $g_1^t, \ldots, g_n^t$ for updating model $x_i$.
> > >
> > > Now we additionally consider using stochastic gradients with batch size $b$ for the inner product. That is, let $h_i^t$ and $h_j^t$ be independent and unbiased estimates of
> > > $\langle\nabla f_i(z_{ij}^t), \nabla f_j(z_{ij}^t)\rangle$. The variance of ${h}_i^t$ has a variance of $\frac{\sigma^2}{b}$.
> > >
> > > Let's denote $\mathbb{E} _ g:=\mathbb{E} _ {g _ 1,\ldots, g _ n}$ and $\mathbb{E} _ h:=\mathbb{E} _ {h _ 1,\ldots, h _ n}$ and replace the expecation $\mathbb{E}$ in the proof with $\mathbb{E} _ h\mathbb{E} _ g$. Notice that the only difference is that
> > >
> > > \begin{align*}
> > >     \mathbb{E} _ h\mathbb{E} _ g\left[\tilde{f} _ {ij}\left(z _ {ij}^{t+1}\right)\right]
> > >     \le& \tilde{f} _ {ij}\left(z _ {ij}^{t}\right) +
> > >     \left\langle \nabla \tilde{f} _ {ij}\left(z _ {ij}^{t}\right), \mathbb{E} _ h\mathbb{E} _ g\left[z _ {ij}^{t+1} - z _ {ij}^{t}\right] \right\rangle + \frac{L}{2} \lVert\mathbb{E} _ h\mathbb{E} _ g[z _ {ij}^{t+1} - z _ {ij}^{t}]\rVert _ 2^2
> > >      + \frac{L}{2} \mathbb{E} _ h\mathbb{E} _ g\left[\lVert z _ {ij}^{t+1} - z _ {ij}^{t} - \mathbb{E} _ h\mathbb{E} _ g[z _ {ij}^{t+1} - z _ {ij}^{t}] \rVert _ 2^2 \right].
> > > \end{align*}
> > >
> > > The last quantity can be expanded as follows:
> > > \begin{align*}
> > >  \mathbb{E} _ h\mathbb{E} _ g \left[\lVert z _ {ij}^{t+1} - z _ {ij}^{t} - \mathbb{E} _ h\mathbb{E} _ g[z _ {ij}^{t+1} - z _ {ij}^{t}] \rVert _ 2^2 \right]
> > > = \mathbb{E} _ h\mathbb{E} _ g \left[\lVert z _ {ij}^{t+1} - z _ {ij}^{t} \pm \mathbb{E} _ h[z _ {ij}^{t+1} - z _ {ij}^{t}] - \mathbb{E} _ h\mathbb{E} _ g[z _ {ij}^{t+1} - z _ {ij}^{t}] \rVert _ 2^2 \right]
> > > =  \mathbb{E} _ h\mathbb{E} _ g \left[\lVert \mathbb{E} _ h[z _ {ij}^{t+1} - z _ {ij}^{t}] - \mathbb{E} _ h\mathbb{E} _ g[z _ {ij}^{t+1} - z _ {ij}^{t}] \rVert _ 2^2 \right]+ \mathbb{E} _ h\mathbb{E} _ g \left[\lVert z _ {ij}^{t+1} - z _ {ij}^{t} - \mathbb{E} _ h[z _ {ij}^{t+1} - z _ {ij}^{t}]  \rVert _ 2^2 \right].
> > > \end{align*}
> > > Plug in the above equality to the above inequality
> > > $$
> > > \begin{align*}
> > >     \mathbb{E} _ h\mathbb{E} _ g\left[\tilde{f} _ {ij}\left(z _ {ij}^{t+1}\right)\right]
> > >     \le& \tilde{f} _ {ij}\left(z _ {ij}^{t}\right) +
> > >     \left\langle \nabla \tilde{f} _ {ij}\left(z _ {ij}^{t}\right), \mathbb{E} _ h\mathbb{E} _ g\left[z _ {ij}^{t+1} - z _ {ij}^{t}\right] \right\rangle + \frac{L}{2} \lVert\mathbb{E} _ h\mathbb{E} _ g[z _ {ij}^{t+1} - z _ {ij}^{t}]\rVert _ 2^2
> > >     + \mathbb{E} _ h\mathbb{E} _ g \left[\lVert \mathbb{E} _ h[z _ {ij}^{t+1} - z _ {ij}^{t}] - \mathbb{E} _ h\mathbb{E} _ g[z _ {ij}^{t+1} - z _ {ij}^{t}] \rVert _ 2^2 \right]
> > >     + \mathbb{E} _ h\mathbb{E} _ g \left[\lVert z _ {ij}^{t+1} - z _ {ij}^{t} - \mathbb{E} _ h[z _ {ij}^{t+1} - z _ {ij}^{t}]  \rVert _ 2^2 \right].
> > > \end{align*}
> > > $$
> > > Among the 5 terms on the right hand side of the above inequality, the first 4 terms is bounded the same way as the original submission. The effect of using stochastic gradient for inner product is limited to
> > > \begin{align*}
> > > \mathbb{E} _ h\mathbb{E} _ g \left[\lVert z _ {ij}^{t+1} - z _ {ij}^{t} - \mathbb{E} _ h[z _ {ij}^{t+1} - z _ {ij}^{t}]  \rVert _ 2^2 \right] \\
> > > =\mathbb{E} _ h\mathbb{E} _ g \left[ \left\lVert \frac{\eta}{2} (g _ i^t + g _ j^t) + \frac{\eta\rho}{2}\sum _ {k=1}^n (w _ {ik}^{t+1} (x _ i^t - x _ k^t) + w _ {jk}^{t+1} (x _ j^t - x _ k^t)) \right.\right.
> > >  \qquad\left.\left. - \mathbb{E} _ h \left[\frac{\eta}{2} (g _ i^t + g _ j^t) + \frac{\eta\rho}{2}\sum _ {k=1}^n (w _ {ik}^{t+1} (x _ i^t - x _ k^t) + w _ {jk}^{t+1} (x _ j^t - x _ k^t)) \right]  \right\rVert _ 2^2 \right]
> > > =\mathbb{E} _ h\left[ \left\lVert \frac{\eta\rho}{2}\sum _ {k=1}^n ((w _ {ik}^{t+1} - \mathbb{E} _ h[w _ {ik}^{t+1}]) (x _ i^t - x _ k^t) + (w _ {jk}^{t+1}- \mathbb{E} _ h[w _ {jk}^{t+1}]) (x _ j^t - x _ k^t))  \right\rVert _ 2^2 \right]
> > > \end{align*}
> > > Then
> > > \begin{align*}
> > > \mathbb{E} _ h\mathbb{E} _ g \left[\lVert z _ {ij}^{t+1} - z _ {ij}^{t} - \mathbb{E} _ h[z _ {ij}^{t+1} - z _ {ij}^{t}]  \rVert _ 2^2 \right] \\
> > > =\frac{\eta^2\rho^2}{4} \sum _ {k\neq i,j} \left(\mathbb{E} _ h \left[ \left\lVert w _ {ik}^{t+1} - \mathbb{E} _ h[w _ {ik}^{t+1}] \right\rVert^2 _ 2 \right]  \lVert x _ i^t - x _ k^t\rVert _ 2^2 + \mathbb{E} _ h \left[ \left\lVert w _ {jk}^{t+1} - \mathbb{E} _ h[w _ {jk}^{t+1}] \right\rVert^2 _ 2 \right]  \lVert x _ j^t - x _ k^t\rVert _ 2^2 \right)
> > > \end{align*}
> > > where we use the independence of random variables in the last equality.

---

> > > > ### Author Response · Authors · 2024-08-12
> > > > **Addressing weakness 1 (Part 2)**
> > > >
> > > > Average over i, j yields
> > > > $$
> > > > \begin{align*}
> > > > \frac{1}{c^2}\sum _ {ij}\mathbb{E} _ h\mathbb{E} _ g \left[\lVert z _ {ij}^{t+1} - z _ {ij}^{t} - \mathbb{E} _ h[z _ {ij}^{t+1} - z _ {ij}^{t}]  \rVert _ 2^2 \right]
> > > > =\frac{\eta^2\rho^2(c-2)}{2c^2} \sum _ {i,j} \mathbb{E} _ h \left[ \left\lVert w _ {ij}^{t+1} - \mathbb{E} _ h[w _ {ij}^{t+1}] \right\rVert^2 _ 2 \right]  \lVert x _ i^t - x _ j^t\rVert _ 2^2  \\
> > > > \le&\frac{\eta^2\rho^2(c-2)}{2c^2} \frac{4\sigma^2}{b} \sum _ {i,j}  \lVert x _ i^t - x _ j^t\rVert _ 2^2.
> > > > \end{align*}
> > > > $$
> > > > Then
> > > > $$
> > > >     \begin{align*}
> > > >         \frac{1}{c^2}\sum _ {i,j\in c}\left\lVert\nabla \tilde{f} _ {ij}\left(z _ {ij}^{t}\right)\right\rVert _ 2^2
> > > >         \le \frac{2}{\eta c^2} \sum _ {i,j\in C}\left( \tilde{f} _ {ij}\left(z _ {ij}^{t}\right) - \tilde{f} _ {ij}\left(z _ {ij}^{t+1}\right)  \right)
> > > >         +  \left(\frac{3L^2}{4c^2} + 3\rho^2 + \frac{4\eta^2\rho^2(c-2)\sigma^2}{b}\right) \sum _ {i,j\in c} \lVert x _ i^t-x _ j^t\rVert _ 2^2
> > > >          +  \frac{3n\rho^2}{c} \sum _ {i\in c} \sum _ {k=1}^n |w _ {ik}^{t+1} - w _ {ik}^\star|^2 \lVert x _ i^t - x _ k^t\rVert _ 2^2 + \frac{L\eta \sigma^2}{2}.
> > > >     \end{align*}
> > > > $$
> > > > Let $\eta \le \sqrt{\frac{b}{(c-2)\sigma^2}}$ we have that $\frac{4\eta^2\rho^2(c-2)\sigma^2}{b} \le 4 \rho^2$. The rest of the proof remain the same except for constants.
> > > >
> > > > Due to character limit and the difficulty of editing long equations in markdown here, we show stop the proof here and note that the rest of the proof remain the same up to constants.

---

> > > > > ### Comment · Reviewer_H7Ja · 2024-08-12
> > > > >
> > > > > Thank the authors for providing more proof. Could the authors provide the final theorem and compare it with Theorem 1 to show the impact of the stochastic gradient and also compare it with stochastic gradient based state-of-the-art algorithms?

---

> > > > > > ### Author Response · Authors · 2024-08-13
> > > > > > **Response**
> > > > > >
> > > > > > The chances are replaced Line 410. By taking $b\ge\frac{16}{3}\eta^2(c-2)\sigma^2$ and $\rho\ge \frac{\sqrt{3}L}{c}$ and $M_{ij}\le\frac{1}{5}$, the following inequality hold true
> > > > > > $$
> > > > > > \begin{align*}
> > > > > >     \frac{3\eta M^2_{ij}}{\rho c} \left(\frac{3L^2}{4} + 3c^2\rho^2 + \frac{4\eta^2\rho^2(c-2)\sigma^2}{b} \right)
> > > > > >     \le \frac{1}{2} \eta\rho c.
> > > > > > \end{align*}
> > > > > > $$
> > > > > > Notice that the upper remain the same. Next is the full theorem where the only change is that we additional require the $b$ is not too small.
> > > > > >
> > > > > > -----
> > > > > >
> > > > > > Suppose Assumption 1,2,3,4,5 hold true. Suppose that Algorithm 1 solves (1) exactly.
> > > > > > Consider clients $i$ and $j$ in the same cluster $C$ of size $c$. Suppose  that $M _ {ij}^2\in(0,\frac{1}{5})$ and $\zeta^2 _ {ik}\ge \lVert \nabla f _ i(x) + \nabla f _ k(x) \rVert _ 2^2$ for all $x$ and $k$.
> > > > > > Let $\rho\ge \frac{\sqrt{3}L}{c}$ and  $\mathbf{b\ge\frac{16}{3}\eta^2(c-2) \sigma^2}$  and step size  is
> > > > > > $$
> > > > > > \begin{align*}
> > > > > >     \eta \le \min \left\\{ \frac{2}{\sigma\sqrt{LT}} \sqrt{ \frac{1}{c^2} \sum _ {i,j\in C}  \left( \tilde{f} _ {ij}\left(z _ {ij}^{0}\right) - \tilde{f} _ {ij}^\star  \right)}, \frac{1}{2\sqrt{3}L}  \right\\}.
> > > > > > \end{align*}
> > > > > > $$
> > > > > > The consensus distance also converges to 0, i.e.
> > > > > > $$
> > > > > > \begin{align*}
> > > > > >     \frac{1}{c^2T}\sum _ {t=0}^{T-1}\sum _ {i,j\in C}  \mathbb{E}\left[\lVert x _ i^{t+1} - x _ j^{t+1}\rVert _ 2^2 \right]
> > > > > >     \le& \frac{6M^2 _ {ij}}{\rho^2 c}
> > > > > >         \sqrt{ \frac{L\sigma^2}{c^2T} \sum _ {i,j\in C} \left( \tilde{f} _ {ij}\left(z _ {ij}^{0}\right) - \tilde{f} _ {ij}^\star  \right)}.
> > > > > > \end{align*}
> > > > > > $$
> > > > > > Moreover, the gradient norm is upper bounded.
> > > > > > $$
> > > > > > \begin{align*}
> > > > > >     \frac{1}{c^2 T}\sum _ {t=0}^{T-1} \sum _ {i,j\in C} \mathbb{E}\left[\lVert\nabla \tilde{f} _ {ij}\left(z _ {ij}^{t}\right)  \rVert _ 2^2\right]
> > > > > >     \le& 3 \sqrt{ \frac{L\sigma^2}{c^2T} \sum _ {i,j\in C} \left( \tilde{f} _ {ij}\left(z _ {ij}^{0}\right) - \tilde{f} _ {ij}^\star  \right)}.
> > > > > > \end{align*}
> > > > > > $$
> > > > > > -----
> > > > > > Regarding the impact of stochastic gradient:  The theoretical results above suggest that as long as $b$ is not too small, that is $\Omega(\eta^2 c)$, the convergence results remain the same. The intuition is that these stochasticity gradients are only used to compute the inner product that approximates the direction $w_{ij}$ should head for. The perturbation may well not change the sign of the inner product. Besides, as $w_{ij}$ is bounded between [0,1], the inner produce of stochastic gradients can be seen as clipped, making it more robust to stochasticity. Throughout the experiments, we indeed use the stochastic gradients variant.
> > > > > >
> > > > > > ----
> > > > > > Regarding the comparison with other stochastic gradient based algorithm: There is no perfect baseline for comparison as our formulation is designed for client-selection and as far as we know different from other works.  But we can refer the rebuttal we posted to Reviewer xPeE that "We provide a comparison of the theoretical convergence performance between CoBo and state-of-the-art algorithms in our response. Specifically, we show that CoBo enjoys linear scalability with respect to cluster size, whereas federated clustering requires O(n^2) gradient computations per iteration. Ditto gives a coarse grained O(1/T) convergence rate for local models."

---

> > > > > > > ### Comment · Reviewer_H7Ja · 2024-08-13
> > > > > > >
> > > > > > > Thank the authors for the response. My concerns have been addressed, and I am willing to increase my score.

---

### Official Review · Reviewer_xPeE · 2024-07-13

**Soundness:** 2
**Presentation:** 3
**Contribution:** 2
**Rating:** 4
**Confidence:** 4

**Summary:**

The paper proposes bi-level training for heterogeneous federated learning, where heterogeneity is due to underlying clustered clients. The two levels of optimizations are model training and determining the client similarity. The authors prove a convergence result and prove empirical results on training vision and LLM models.

**Strengths:**

- Paper is very easy to follow and understand due to the flow of language.
- The presented results indicate the advantages compared to other works.
- The high level idea is sound.

**Weaknesses:**

- For each experiment the setting is fixed, i.e. there are no ablations on how e.g. number of clusters, data samples per client, number of clients per cluster affects the performance. There is a sparsity in experiments.
- FedAvg+fine-tuning should be compared to as an additional method.
- The convergence result on f_ij instead of on f_i, I think this should be fixed since the optimization problem is on f_i's.
- The algorithm requires pairwise computations across clients which might be huge overhead for practical applications where there might be millions of clients.
- More discussion is needed for the theorem, e.g. what does it shows us, how are different setting parameters like number of clusters, clients per clusters etc. affect the convergence.

**Questions:**

- What does "each cluster clients share stationary points" mean?
- Why is the convergence result is given on f_ij
- What is the performance of method when clustering assumption does not hold, for instance there is only one cluster?

**Limitations:**

Yes

---

> ### Author Rebuttal · Authors · 2024-08-07
>
> We thank the reviewer for their thoughtful feedback on our paper. We address each of the comments below:
> 1. Sparsity in experiments and ablations:
> We have added more experiments to investigate the effect of different settings, such as the number of clusters, data samples per client, and number of clients per cluster, on the performance of our method. This provides a more comprehensive understanding of the robustness and generalizability of CoBo. The full experiment is available in the global PDF file. As expected, the total amount of data available to each cluster has a direct influence on the performance of the method. Additionally in the experiments, CoBo is almost invariant to the number of clusters. In the experiment for different number of clients per cluster, the dataset is partitioned among clients in each cluster, meaning that larger cluster sizes result in less data available to each individual client. Nevertheless, CoBo can utilize collaboration to preserve the performance for larger cluster sizes, and remain invariant to cluster size as well.
> 2. Comparison to FedAvg+fine-tuning:
> We have included a comparison of our method with FedAvg+fine-tuning in the revised manuscript, allowing for a more complete evaluation of the performance of CoBo relative to existing methods. The performance of this baseline could be found in the table available in the global PDF file. We confirm that CoBo can outperform this method in all experiments.
> 3. Convergence result on f_i:
> We agree with the reviewer that the results should be given for $f_i$. We note that the convergence results for $f_i$ can be easily derived by combining the last equation of Theorem I with Equation (5). That is, by adding
> $\lVert \nabla f_i(x) + \nabla f_j(x) \rVert_2^2$ to both sides of the Equation (5), we have that
> $$2(\lVert \nabla f_i(z_{ij}) \rVert_2^2 + \lVert \nabla f_j(z_{ij}) \rVert_2^2) \le 4 (1+ M_{ij}^2)  \lVert \nabla  f_i (z_{ij})  \rVert_2^2 $$
> Then using the upper bound of $M_{ij} < ⅕$ in Theorem 1 and average over t and i,j yields
> $$\frac{1}{c^2T} \sum_{t=0}^{T-1} \sum_{ij\in c} \lVert \nabla f_i(z_{ij}) \rVert_2^2 \le (1+1/25) RHS$$
> where RHS refers to the right hand side of the last equation of Theorem 1. The convergence of $\lVert \nabla f_i(x_i) \rVert_2^2$ can also be derived using Cauchy-Schwarz inequality and apply the consensus distance inequality in Theorem 1.
> 4. Pairwise computation overhead:
> We agree with the reviewer that the pairwise gradient computation in the inner problem can be expansive. In fact, we had already addressed this issue in Section 2.1 of the manuscript by uniformly sampling only O(1/n) percent of edges in each iteration, leaving the computation complexity of the inner problem the same order as the outer problem. We also included an experiment to empirically compare different sampling strategies, available in the global PDF file. The experiment demonstrates that CoBo is robust to different sampling strategies, while proposing a sampling method that slightly increases the performance.
>
> 5. Discussion of the theorem:
> We thank the reviewer for the insightful suggestion. We will incorporate the discussions into the paper. As M_ij measures how well i,j collaborate, in fact, smaller M_{ij} leads to better consensus distance, with M_ij=0 leading to always identical as expected. The gradient norms does not scale linearly with the number of clients because we are considering the norm of individual gradients. (This scaling will appear when we consider the averaged gradient among all clients in this cluster. We will add the proof in the future version.)
>
>
>
> Regarding questions:
> 1. Clients sharing stationary points:
> The statement "each cluster clients share stationary points" means that, despite having different data distributions, the clients within a cluster can simultaneously reach a stationary point. This assumption is indeed a relaxation of the i.i.d. assumption, allowing for more realistic and heterogeneous data distributions.
> 3. Performance without clustering assumption:
> Our algorithm still applies when there is only one cluster as it does not require knowledge of the exact number of clusters or balanced clusters.
>
> We hope that our revisions have  addressed the concerns raised by the reviewer, and increase the score if possible.

---

> > ### Comment · Reviewer_xPeE · 2024-08-10
> >
> > Thanks for the rebuttal, comparison to FedAvg+ft is a good addition. Point 3. is very hard to follow as you have not utilized markdown. Despite sampling, I still think the pairwise computations are an important overhead in practice; also the theoretical results does not take into account such a sampling occurs. Hence, I would like to keep my score.

---

> > > ### Author Response · Authors · 2024-08-12
> > > **Response**
> > >
> > > We are glad to have addressed your concern about FedAvg+Fine-tuning. Regarding the other points, we would like to clarify that:
> > > - CoBo, as shown in Algorithm 1, **only computes O(n) gradients per iteration**, instead of pairwise O(n^2), making it very efficient in practice. In a setup with n clients, if we sample the pairs of clients with probability O(1/n), the expected number of selected pairs in each timestep is n, which is the same as the number of local computed gradients in all baselines. Hence, the complexity of our algorithm is the same as other collaborative learning algorithms. As is shown in the experiments sections of the submission pdf and rebuttal pdf, CoBo is able to reach **higher accuracies** compared to other methods **in all experiments** with **same order of complexity**.
> > >
> > > - As for point 3, we are not sure why the reviewer says that “we have not utilized markdown” as the equations render correctly to us. Could you try again or try other browsers?
> > >
> > > - As for theoretical results, the technical challenges of the convergence proof are already addressed in the no-sampling theorem while the sampling variant is a simple corollary. We will add the simple corollary to the main text.
> > >
> > > We hope that our responses clear the reviewer’s main concern about complexity.

---

### Author Rebuttal · Authors · 2024-08-07

We thank the reviewers for their insightful reviews and feedback. **The attached PDF**, contains additional experiments allowing us to gladly address the following concerns:

**1. FedAvg+fine-tuning:** A new baseline is included for all experiments. This baseline is similar to FedAvg for the first 80\% of the training, and then it fine-tunes each client on their local dataset for the remainder of the iterations. Although this method performs better than FedAvg, CoBo's performance is still superior in all experiments.

**2. Ablation study on the Cross-silo experiment:** We have added new experiments to see how CoBo behaves with different experimental setups. As expected, CoBo is sensitive to the total data available to each cluster. We also found out that increasing the number of clusters does not affect the performance of CoBo significantly. We also experimented with the different number of clients per clusters, maintaining the fixed number of 4 clusters. Since the data is partitioned among clients of each cluster, larger cluster size means lower data available to each individual client. However, we are glad that CoBo can also preserve the accuracy for larger clusters, by enabling collaboration within clusters.

**3. Sampling strategies for updating collaboration matrix:** We agree that the sampling method is a very important part of our algorithm. We therefore conduct experiments on different sampling strategies in the Cross-silo experiment. Our proposed methods almost perform similarly, suggesting the robustness of CoBo on its sampling method. We observed that CoBo can find collaborators in the early stages of the training, leading us to propose a new sampling method based on sampling more frequently in the early stages and reduce the frequency in the later stages. We show that this mixed strategy can slightly perform better than other strategies in the Cross-silo experiment.

**4. Repeated Experiments:** We acknowledge the importance of the repeated experiments to examine the robustness of methods, and we thank the reviewers for raising this point. We have added the confidence intervals for the Cross-device experiment as well in order to address this matter.

---

### Decision · Program_Chairs · 2024-09-25

**Decision:**

Accept (poster)

**Comment:**

This paper introduces a new approach to collaborative learning by framing it as a bilevel optimization problem, where the lower-level optimization problem aims to learn the communication topology, and the upper-level problem aims to train machine learning models. The authors provide a convergence analysis for the proposed method, and the experimental results confirm its efficacy.

Strengths:
1. Formulating collaborative learning as a bilevel optimization problem is innovative and allows for learning a problem-adaptive topology.
2. The paper includes a theoretical analysis of the proposed method, and the experimental results confirm its efficacy.

However, there are some minor concerns regarding the convergence analysis. In particular, the theoretical analysis in the original submission assumes the use of the full gradient in the lower-level optimization problem, which is inconsistent with Algorithm 1 and the experiments. During the rebuttal phase, the authors addressed this issue by providing a new analysis based on the stochastic gradient.

Overall, the idea is interesting, and the concerns regarding the convergence analysis have been addressed. Therefore, I suggest that the authors revise the paper as the reviewer suggested and recommend accepting it.